# Communicating Activations Between Language Model Agents

**Vignav Ramesh** [1]   **Kenneth Li** [1]

## Abstract

Communication between multiple language model (LM) agents has been shown to scale up the reasoning ability of LMs. While natural language has been the dominant medium for inter-LM communication, it is not obvious this should be the standard: not only does natural language communication incur high inference costs that scale quickly with the number of both agents and messages, but also the decoding process abstracts away too much rich information that could be otherwise accessed from the internal activations. In this work, we propose a simple technique whereby LMs communicate via *activations*; concretely, we pause an LM $B$'s computation at an intermediate layer, combine its current activation with another LM $A$'s intermediate activation via some function $f$, then pass $f$'s output into the next layer of $B$ and continue the forward pass till decoding is complete. This approach scales up LMs on new tasks with *zero* additional parameters and data, and saves a *substantial amount of compute* over natural language communication. We test our method with various functional forms $f$ on two experimental setups—multi-player coordination games and reasoning benchmarks—and find that it achieves up to $27.0\%$ improvement over natural language communication across datasets with $<1/4$ the compute, illustrating the superiority and robustness of activations as an alternative "language" for communication between LMs.

## 1. Introduction

Language is for the purpose of communication. As large language models (LLMs) have been increasingly used to power autonomous, goal-driven agents capable of reasoning, tool usage, and adaptive decision-making (Yao et al.,

---
[1]Kempner Institute for AI, Harvard University, Cambridge, MA, USA. Correspondence to: Vignav Ramesh <vignavramesh@college.harvard.edu>.

*Proceedings of the 42$^{nd}$ International Conference on Machine Learning*, Vancouver, Canada. PMLR 267, 2025. Copyright 2025 by the author(s).

2023; Xi et al., 2023; Wang et al., 2024; Ahn et al., 2022; Schick et al., 2023; Shen et al., 2023; Park et al., 2023; Nakano et al., 2022), communication between multiple co-operating agents has emerged as an intuitive approach to amplify the reasoning capabilities of LLMs (Wu et al., 2023). Explicit communication in natural language between multiple LLMs has been shown to encourage divergent thinking (Liang et al., 2023), improve factuality and reasoning (Du et al., 2023), enable integration of cross-domain knowledge (Sukhbaatar et al., 2024), and allow for modular composition of abilities in a complementary manner (Wu et al., 2023; Prasad et al., 2023).

A critical problem with natural language communication, however, is that it incurs extremely high inference costs that scale quickly with the number of agents as well as length and number of messages (Du et al., 2023; Yang et al., 2023; Wu et al., 2023). Restricting LLM communication to natural language also raises the question: as LLMs are increasingly capable of handling larger, more complex tasks (sometimes with "super-human" ability) (Wei et al., 2022; Burns et al., 2023), might they communicate more effectively in representations of higher dimension than natural language? While using natural language as a communicative medium is appealing due to its interpretability, we claim that it may not be optimal for inter-LLM communication. Natural language generation uses only one token to represent the model's belief over the entire vocabulary, which risks losing information embedded within the model output logits (Pham et al., 2024); furthermore, a model's belief over the entire vocabulary is itself not always better (for communicative purposes) than the model's (often richer) representation of the input in earlier layers. Indeed, Hernandez et al. (2024) find that by around the halfway point of an LM's computation, it has developed "enriched entity representations" of the input, where entities in the prompt are populated with additional facts about that entity encoded in the model's weights; but by the later layers these embeddings are transformed into a representation of the next word which leverages only parts of the previous, richer representations, when that full embedding would be quite useful for communication.

Motivated by these concerns, this work outlines a simple technique whereby LLM agents communicate via *activations*, thus enabling more efficient (i.e., higher-entropy) com-

munication at a fraction of the number of forward passes required at inference time. Concretely, we (1) pause a Transformer LM $B$'s computation at intermediate layer $j$ in the residual stream; (2) combine its post-layer $j$ activation with another LM $A$'s post-layer $k$ activation via some function $f$; and then (3) pass $f$'s output into the next layer $j + 1$ of $B$ and continue its forward pass till decoding is complete. This approach scales up LLMs on new tasks by leveraging existing, frozen LLMs along with *zero* task-specific parameters and data, applying to diverse domains and settings. Furthermore, in requiring only a partial forward pass through $A$ and one forward pass through $B$, this method saves a *substantial amount of compute* over traditional natural language communication, which we quantify in Section 3.2.

We validate our method by testing this approach with various functional forms $f$ on two experimental setups: two multiplayer coordination games, where $B$ is asked to complete a task requiring information provided in a prompt to $A$; and seven reasoning benchmarks spanning multiple domains: Biographies (Du et al., 2023), GSM8k (Cobbe et al., 2021), MMLU High School Psychology, MMLU Formal Logic, MMLU College Biology, MMLU Professional Law, and MMLU Public Relations (Hendrycks et al., 2021). Our activation communication protocol exhibits up to 27.0% improvement over natural language communication across these datasets, using $<1/4$ the compute. Critically, unlike prior work which test inter-LLM communication only on large-scale ($>$70B) models (Du et al., 2023; Liang et al., 2023), we find that our approach generalizes across a wide array of LLM suites and sizes, enabling even smaller LLMs to unlock the benefits of communication.

In summary, our contributions are two-fold:

- We propose a novel inter-model communication protocol for LLM agents that is purely activation-based.

- We perform comprehensive experiments to validate the improved performance of activation communication over traditional natural language communication. We also formally quantify our approach's compute savings over natural language communication, illustrating the superiority and robustness of activations as an alternative "language" for communication between LMs.

## 2. Related Work

**Multi-agent communication** The field of multi-agent communication has a long-standing history. Notably, prior works on emergent communication have showed that agents can autonomously evolve communication protocols when deployed in multi-agent environments that enable cooperative and competitive game-play (Sukhbaatar et al., 2016; Foerster et al., 2016; Lazaridou et al., 2017). However, recent experiments have demonstrated that learning meaning-

ful languages from scratch, even with centralized training, remains difficult (Lowe et al., 2020; Chaabouni et al., 2019; Jaques et al., 2019).

With the emergence of large pre-trained language models, allowing communication between LLMs in natural language has hence become a promising approach to enable coordination among multiple LLM agents (Li et al., 2023). Recent works have demonstrated that such conversations enable integration of cross-domain knowledge (Sukhbaatar et al., 2024), modular composition of abilities in a complementary manner (Wu et al., 2023), and improved task performance via splitting into subtasks (Prasad et al., 2023). Most notable is multiagent debate introduced by Du et al. (2023), where LLMs provide initial responses and then make refinements by iteratively considering inputs from peers. While such methods have been shown to improve performance on various tasks over vanilla and majority-vote (Wang et al., 2023) style prompting, these experiments have only focused on large models (GPT-3.5/4, LLaMA2-70B and up), leaving the efficacy of debate on smaller, open-source models underexplored; our study addresses this gap by reimplementing Du et al. (2023) in experiments with smaller-scale ($1 - 70$B) models. More crucially, debate and similar natural language communication methods are *extremely computationally expensive*, which this work addresses (Yang et al., 2023; Wu et al., 2023).

Notably, Pham et al. (2024) propose CIPHER, which uses *input (tokenizer) embeddings* (as opposed to activations) to enable multi-agent communication; specifically, CIPHER passes the average tokenizer embedding (weighted by the LLM's next-token probabilities) between models. While (Pham et al., 2024) show this approach outperforms natural language debate, it (i) still faces substantial information loss relative to the model *activations* and (ii) does not save compute, as the number of these "average embeddings" passed between models is the same as the number of tokens passed between models in natural language communication.

A related class of methods involves spending extra test-time compute reasoning in latent space (Geiping et al., 2025; Hao et al., 2024). Such latent reasoning approaches involving doing "chain-of-thought in activation space," e.g. by grafting LM activations into other layers/later forward passes through the same model (e.g., a form of "recurrent AC" within a single model); our approach can be viewed as doing exactly the same thing, but instead "outsourcing" the CoT to another model (and thus reaping benefits from greater diversity of thoughts/reasoning paths from distinct models).

**Activation engineering** Activation engineering involves editing an LLM's intermediate layer representations during a forward pass to create desired changes to output text (Li

et al., 2024; Turner et al., 2023). Past work has explored extracting latent steering vectors from a frozen LLM to control quality and content of completions (Subramani et al., 2022), as well as using "direction" vectors (computed as the difference in activations between two prompts) that enable inference-time control over high-level properties of generations (Li et al., 2024; Turner et al., 2023). This work involves activation editing that is similar to such prior works at a high level, though for the purpose of communication between LLM agents.

**Model composition and grafting**  Composing expert models has been a recurring strategy to improve large models, with different methods imposing different restrictions on the types of base LLMs that can be combined. Mixture of Experts (Shazeer et al., 2017) requires that all experts are trained simultaneously using the same data; Branch-Train-Mix (Sukhbaatar et al., 2024) trains a single base LM multiple times on different datasets, then learns a router on outputs. Crucially, these methods do not work when neither model can do the task at hand well (i.e., they solve the problem of choosing which of several outputs is best, not that of generating a high-quality output by recombining the disparate abilities of the various base LMs).

Model grafting, in contrast, seeks to merge different models immediately prior to or at inference-time. Past works have explored this at the parameter level (e.g., task vector averaging as in Ilharco et al. (2023), which requires that the base models be well aligned), probability distribution / token level as in Shen et al. (2024) (which imposes few restrictions on the relationship between the base models, but by virtue of being token-based can result in cascading errors during decoding), and activation level (e.g., CALM (Bansal et al., 2024) which learns an attention layer on top of two models' intermediate layer activations and thus enables broader integration of model abilities than token-level methods, but requires re-tuning of the attention mechanism for every model pair). In this work, we seek to unify CALM and other activation-level grafting techniques under a single framework, parameterized by the function $f$ used to combine activations; crucially, we explore simple forms of $f$ (e.g., sum, mean) that—unlike Bansal et al. (2024)—require *zero additional task-specific parameters and data*, and are far more compute-efficient.

## 3. Communicating Activations Between Language Models

We propose a simple yet effective technique whereby language models communicate via *activations*. We detail our approach in Section 3.1; provide analytical models of the compute saved over natural language communication in Section 3.2; and discuss the intuition behind this approach in

Section 3.3.

### 3.1. Method

Consider two language models, $A$ and $B$, and some setting in which $B$ must perform a task where it would benefit from knowledge given to $A$ as a prompt/encoded in $A$'s weights (example settings in Section 4.1/Section 4.2 respectively). We propose incorporating information from $A$'s post-layer $k$ activation $\boldsymbol{h}_{A,k}$ into $B$'s post-layer $j$ activation $\boldsymbol{h}_{B,j}$ (and vice versa, though for simplicity we henceforth only discuss the first direction) (Figure 1, left).

More formally, suppose $A$ and $B$ (which have model dimensions $d_A$ and $d_B$ respectively) are given prompts $x_A$ and $x_B$ respectively, where $x_A$ is of length $t_A$ tokens and $x_B$ is of length $t_B$ tokens. We first run a partial forward pass of $B$ until layer $j$ (henceforth denoted $B_{\leq j}(x_B)$) to get $\boldsymbol{h}_{B,j} \in \mathbb{R}^{t_B \times d_B}$. Then we (1) run a partial forward pass of $A$ until layer $k$ to get $A_{\leq k}(x_1) := \boldsymbol{h}_{A,k} \in \mathbb{R}^{t_A \times d_A}$; (2) replace the activation of the last token $(\boldsymbol{h}_{B,j})_{t_B} \in \mathbb{R}^{d_B} \longleftarrow f((\boldsymbol{h}_{A,k})_{t_A}, (\boldsymbol{h}_{B,j})_{t_B})$ for some function $f : \mathbb{R}^{d_A + d_B} \rightarrow \mathbb{R}^{d_B}$; then (3) continue $B$'s forward pass till decoding is complete, resulting in an output $y = B_{>k}(\boldsymbol{h}_{B,j})$.

Let $\boldsymbol{a} = (\boldsymbol{h}_{A,k})_{t_A}$, $\boldsymbol{b} = (\boldsymbol{h}_{B,j})_{t_B}$. For sake of simplicity assume $d_A = d_B$.[1] We consider three non-learned functions $f$:

$$f(\boldsymbol{a}, \boldsymbol{b}) = \boldsymbol{a} + \boldsymbol{b} \qquad (\texttt{sum})$$

$$f(\boldsymbol{a}, \boldsymbol{b}) = \frac{1}{2}(\boldsymbol{a} + \boldsymbol{b}) \qquad (\texttt{mean})$$

$$f(\boldsymbol{a}, \boldsymbol{b}) = \boldsymbol{a} \qquad (\texttt{replace})$$

For cases where, due to differences in $A$ and $B$'s training, $A$ and $B$'s activation spaces are quite different, we propose learning a *task-agnostic* (depends only on the models $A$ and $B$) linear layer $\boldsymbol{W} \in \mathbb{R}^{d_B} \times \mathbb{R}^{d_A}$ that projects $\boldsymbol{a}$ onto $B$'s activation space. Note that this introduces zero additional task-specific parameters and data, as we propose learning this "mapping matrix" $\boldsymbol{W}$ only once for each model pair $(A, B)$ using general text, e.g. sequences from $A$ and/or $B$'s pretraining data mixes. We can then perform sum, mean, or replace with $\boldsymbol{W}\boldsymbol{a}, \boldsymbol{b}$ instead of $\boldsymbol{a}, \boldsymbol{b}$. We propose training $\boldsymbol{W}$ to minimize MSE loss over a dataset of $N$ sentences

$$\mathcal{L}_{\text{MSE}}\left(\{\boldsymbol{y}^{(i)}\}_{i=1}^{N}, \{\boldsymbol{z}^{(i)}\}_{i=1}^{N}\right) = \frac{1}{N}\sum_{i=1}^{N}\left\|\boldsymbol{z}^{(i)} - \boldsymbol{W}\boldsymbol{y}^{(i)}\right\|_2^2$$

---

[1]When $d_A \neq d_B$, the sum, mean, and replace functions are defined as follows. Let $d = \min(d_A, d_B)$ and $\circ$ the concatenation operator. Then $f(\boldsymbol{a}, \boldsymbol{b}) = \boldsymbol{b}_{1:\max(d_B-d,0)} \circ \left(\boldsymbol{b}_{\max(d_B-d,0)+1:d_B} + \boldsymbol{a}_{\max(d_A-d,0)+1:d_A}\right)$ (sum), $f(\boldsymbol{a}, \boldsymbol{b}) = \boldsymbol{b}_{1:\max(d_B-d,0)} \circ \frac{1}{2}\left(\boldsymbol{b}_{\max(d_B-d,0)+1:d_B} + \boldsymbol{a}_{\max(d_A-d,0)+1:d_A}\right)$ (mean), and $f(\boldsymbol{a}, \boldsymbol{b}) = \boldsymbol{b}_{1:\max(d_B-d,0)} \circ \boldsymbol{a}_{\max(d_A-d,0)+1:d_A}$ (replace).

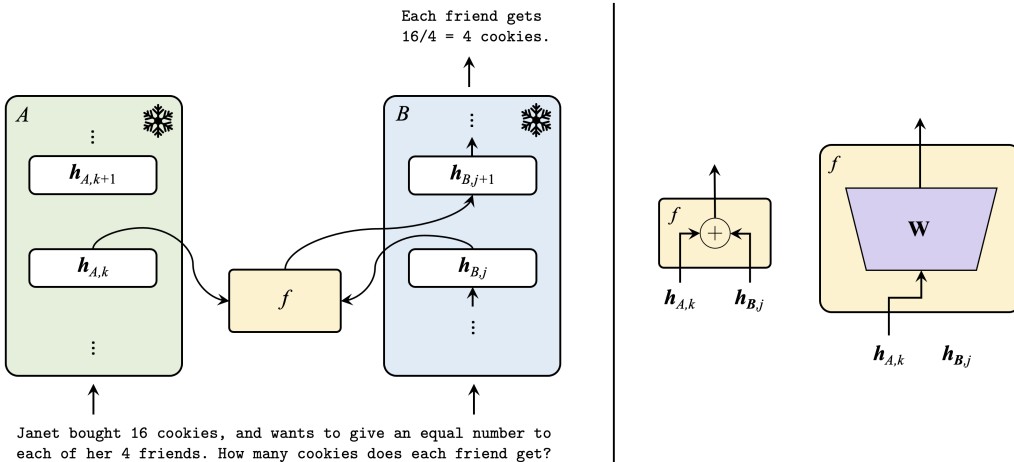

*Figure 1.* **Overview of activation communication.** (Left) Our method involves (1) pausing a Transformer LM $B$ 's computation at layer $j$ in the residual stream; (2) combining its post-layer $j$ activation with another LM $A$ 's post-layer $k$ activation via some function $f$ ; then (3) passing $f$ 's output into the next layer $j + 1$ of $B$ and continuing the forward pass till decoding is complete. (Right) Any function $f$ can be used to combine $A$ and $B$'s activations; we explore letting $f$ be the sum, mean, and replacement functions, as well as a task-agnostic learned linear layer (details in Section 3.1).

where each $(\boldsymbol{y}^{(i)}, \boldsymbol{z}^{(i)})$ pair denotes the final-token layer-26 activations of $A$ and $B$ at layers $k$ and $j$ respectively given the same sentence as input.

## 3.2. Compute Analysis

To understand the significance of activation communication, we must formally quantify the compute this procedure saves over natural language communication. For simplicity suppose the following (similar calculations can be made for the cases where $A$ and $B$ have differing model architectures and/or are given different prompts):

- $A$ and $B$ both have $L$ layers (each with $H$ attention heads, key size $K$, and feedforward size $F$), dimension $D$, and vocab size $V$
- $A$ and $B$ are both given a prompt of $P$ tokens
- $A$ can send $B$ a single $M$-token message
- $B$ must produce an output of $T$ tokens, given its prompt and $A$'s message

Traditional methods require $M$ forward passes of $A$ given a $P$-length input, plus $T$ forward passes of $B$ given a $(P+M)$-length input. Following Hoffmann et al. (2022), this requires

$$
\begin{aligned}
& M\big(4PVD + L(8PDKH + 4P^2KH + 3HP^2 \\
& + 4PDF)\big) + T\big(4(P + M)VD + L(8(P + M)DKH \quad (1) \\
& + 4(P + M)^2KH + 3H(P + M)^2 + 4(P + M)DF)\big)
\end{aligned}
$$

FLOPs. In contrast, at inference time, our method requires only 1 partial (up till the $k$th layer) forward pass of $A$ given

a $P$-length input, $T$ forward passes of $B$ given a $P$-length input, and the activation replacement procedure. This requires

$$
\begin{aligned}
& 2PVD + k(8PDKH + 4P^2KH + 3HP^2 \\
& + 4PDF) + T\big(4PVD + L(8PDKH + 4P^2KH \quad (2) \\
& + 3HP^2 + 4PDF)\big) + \mathcal{F}(D)
\end{aligned}
$$

FLOPs, where $\mathcal{F}(D) = O(D)$ for non-learned $f$ and $O(D^2)$ when $f$ is the mapping matrix.

In all practical cases, (2) is *substantially* lower than (1).

## 3.3. Why should this work?

Recall that Pham et al. (2024) propose CIPHER—communicating the average tokenizer embedding (weighted by the LLM's next-token probabilities) between models. We build upon the intuition behind CIPHER, which goes as follows: the token sampling process during decoding risks substantial information loss from the model's output logits, and communicating a model's weighted-average tokenizer embedding essentially entails communicating both that model's final answer and its belief in that answer (over the entire vocabulary).

Communicating activations, then, can be thought of as communicating a strict superset of {next-token prediction, belief over entire vocabulary}, as activations of late-enough layers essentially encode the model's entire knowledge about the provided context as well as its predicted completion and confidence in that completion (see Figures 1 and 7 in Hewitt & Manning (2019) and Hernandez et al. (2024),

respectively, which show that linear probes tasked with predicting certain output characteristics from a Transformer's intermediate layer embeddings of its input work poorly for early layers, extremely well after around the halfway point of computation, but then probe accuracy drops closer to the final layers).[2] Indeed, these curves of probe accuracy by layer indicate that the final layers and LM head "*throw away*" information not useful for next-token prediction that very well could be useful for communicative purposes; this is precisely why our proposed activation communication technique is not an iterative approach (there is no notion of "rounds" like in debate and CIPHER, which require an additional token budget to extract more and more information out of the LM), as one activation grafting step from $A$ to $B$ inherently communicates to $B$ all of $A$'s knowledge/beliefs about the prompt it was given. Moreover, the extra information over the model's next-token prediction and confidence that is encoded in its activations is what makes activation communication more performant than its natural language counterpart, as we will see in Section 4.

## 4. Experiments

We test our method on two distinct experimental setups: multi-player coordination games (Section 4.1) and reasoning benchmarks (Section 4.2). Qualitative results are available in Appendix A.

### 4.1. Multi-player coordination games

Drawing from existing literature on multi-agent communication, we design two Lewis signaling games (Lewis, 2008; Lazaridou et al., 2016) to test the efficacy of activation communication (example prompts and answers in Table 1):

1. **Countries**, where $A$ is given as input a string of the format "[PERSON] is at the [LANDMARK]" and $B$ is asked "Which country is [PERSON] located in?"

2. **Tip Sheets** (inspired by Lewis et al. (2017)), where $A$ is given a simulated "tip sheet" and $B$ is asked to make an informed investment decision in accordance with the information in the tip sheet.

We synthetically generate 100 (**Countries**) and 70 (**Tip Sheets**) different prompts and answers of the same format as the samples in Table 1, and report the proportion out of those samples that $B$ responds with an exact string match to the ground truth answer. As baselines, we consider a "silent"

---

[2]Note one important critique of multiagent debate: that in cases where multiple agents are uncertain about the answer, there is no reason why referencing other agents' answers would generate more factual reasoning. Both CIPHER and activation communication solve this problem, as some notion of model confidence is being communicated along with its next-token prediction.

(✗) setup, where the agents are not allowed to communicate; a "single-agent skyline," where a single LLM is given the concatenation of $A$ and $B$'s prompts; and traditional natural language communication, where $A$ is asked to output a message that is then given to $B$ along with $x_B$. All decoding is done greedily.

Table 2 presents the results for both coordination games using 2 different instances of the same model as the agents ($A = B$). Across the 3B and 8B model sizes, activation communication (AC) with $f = $ replace *almost completely recovers* the gap between the zero-communication (✗) and the single-agent skyline (SKYLINE), *outperforming* natural language communication (NL) using far less compute. We hypothesize that replace is more effective than mean and sum as the former is guaranteed to output a vector within $B$'s activation space, while the latter two likely do not (e.g., the norm of the vector outputted by sum will be around double that of a typical activation). Furthermore, most of the information $B$ needs is likely contained in its representations of previous tokens in the sequence, hence losing its final-token representation does not hurt.

### 4.2. Reasoning Benchmarks

Next, we test our methods on a variety of reasoning benchmarks, spanning several real-world tasks and domains.

**Baselines** We benchmark activation communication against the following two baselines:

- **Single Model**: A single LLM responds to the prompt in natural language.

- **Natural Language Debate (NLD)** (Du et al., 2023): Each LLM provides an initial response to the given prompt. Then, for each of $r - 1$ subsequent rounds, each LLM is prompted to refine its previous response given the other agents' responses as input. Note that NLD is the most direct baseline for our approach, as it is a state-of-the-art natural language communication protocol. We fix $r = 2$ in our experiments.

Note that we do not compare to Pham et al. (2024), as they communicate the *input* (tokenizer) embeddings rather than activations/output embeddings between models, and hence require a shared tokenizer and embedding table between agents which is extremely restrictive and prevents applicability to our experimental setup.

To determine the values of $k$ and $j$ for activation communication (AC), we compute the accuracy on Countries and Tip Sheets for every pair $(k, j) \in \{1, \ldots, 30\}^2$. Based on these results (shown in Figure 2) as well as Table 2, we fix $k = j = 26$ and $f = $ replace for the following experiments.

Table 1: **Multi-player coordination games.** Sample (prompt, answer) pairs for each game.

| Game | Sample Prompts & Ground-Truth Answer |
|---|---|
| Countries | $x_A$: "Alice is at the Acropolis of Athens." 
 $x_B$: "Which country is Alice located in?" 
 *B's Expected Answer*: "Greece" |
| Tip Sheets | $x_A$: "Acme Inc. has taken a nosedive, as its quarterly earnings have dipped 8%. Meanwhile Doe LLC and Kiteflyer Labs have both reached record-high stock prices of 89, but Kiteflyer is involved in an IP lawsuit with its competitors." 
 $x_B$: "You must invest in one company out of {Acme Inc., Doe LLC, Kiteflyer Labs}. Which do you invest in?" 
 *B's Expected Answer:* "Doe LLC" |

Table 2: **Accuracies (%) on both coordination games using two identical** LLaMA **family models.** Communication at layer $k = j = 26$. 95% confidence intervals (1000 bootstrap iterations) reported in parentheses.

| Model | Method | Accuracy (Countries) | Accuracy (Tip Sheets) |
|---|---|---|---|
| LLaMA-3.2-3B | ✗ | 0.0 (0.0, 0.0) | 38.6 (38.6, 39.4) |
| | Skyline | 84.0 (83.5, 84.1) | 100.0 (100.0, 100.0) |
| | NL | 69.0 (68.7, 69.3) | 74.3 (74.0, 74.6) |
| | AC (sum) | 34.0 (33.9, 34.4) | 50.0 (49.6, 50.3) |
| | AC (mean) | 36.0 (35.5, 36.1) | 80.0 (79.8, 80.4) |
| | AC (replace) | **78.0** (77.7, 78.2) | **90.0** (89.9, 90.3) |
| LLaMA-3.1-8B | ✗ | 2.0 (1.9, 2.1) | 54.3 (54.2, 54.5) |
| | Skyline | 86.0 (85.7, 86.1) | 100.0 (100.0, 100.0) |
| | NL | 77.0 (76.6, 77.1) | 85.7 (85.3, 85.8) |
| | AC (sum) | 71.0 (70.9, 71.4) | 85.7 (85.5, 86.0) |
| | AC (mean) | 70.0 (69.7, 70.3) | 92.9 (92.7, 93.1) |
| | AC (replace) | **83.0** (82.7, 83.1) | **95.7** (95.6, 95.9) |

Across all experiment configurations, we fix the decoding strategy to nucleus sampling with $p = 0.9$.

**Models** We conduct most of our experiments using LLaMA-3.2-3B and LLaMA-3.1-8B as the two agents. Additionally, to test our approach's robustness and generalizability, we conduct experiments with models belonging to various other suites within the LLaMA family and of several different sizes.

Note that for these experiments, we restrict the setting to communication between *different* models (rather than multiple instances of the same model in Section 4.1), since the same model would have identical activations for the same prompts, meaning no information would be communicated in the grafting process. We argue that the multiple-model setting is realistic (perhaps more so than the setting of multiple instances of the same model), as recent advances in LLM development have led to the release of models with specialized abilities (Singhal et al., 2023) and of different sizes (Dubey et al., 2024) that merit complementary usage. Our work thus answers the question: *How can we get the best performance by leveraging multiple models of distinct capabilities and sizes, relative to the added inference-time compute over a single forward pass through any single model?*

**Datasets** We evaluate our technique on seven reasoning datasets that span various real-world tasks and domains: (i) **Biographies** (Du et al., 2023), which asks the LLM to generate a factual biography of a famous computer scientist; (ii) **GSM8k** (Cobbe et al., 2021), a variety of grade school math problems created by human problem writers; and (iii) 5 datasets randomly drawn from MMLU (Hendrycks et al., 2021): **High School Psychology** (from the Social Sciences category), **Formal Logic** (from the Humanities category), **College Biology** (from the STEM category), **Professional Law** (from the Humanities Category), and **Public Relations** (from the Social Sciences category). We evaluate on a randomly-sampled size-100 subset of each dataset.

In experiments involving the mapping matrix $W$, we instantiate $W \in \mathbb{R}^{4096 \times 3072}$ using Xavier initialization and

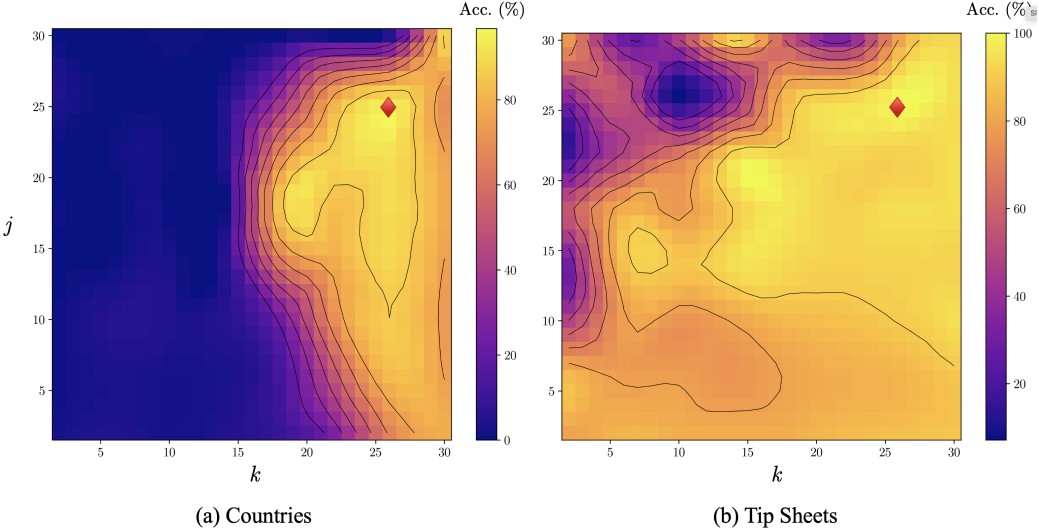

(a) Countries

(b) Tip Sheets

*Figure 2.* **2D contour plots of accuracy over different values of** $k$ **and** $j$ **(the layers at which we access/edit activations for** $A$/$B$ **respectively).** $k = j = 26$ is roughly optimal (◆) for both (a) Countries and (b) Tip Sheets.

train for 10 epochs on a dataset of 3072 sentences[3] randomly drawn from the Colossal Clean Crawled Corpus (C4) (Dodge et al., 2021). We use batch size 32 and the Adam optimizer with learning rate 0.001.

**Metrics**   We measure the accuracy of the final response for the single models and AC. For NLD, we measure the accuracy of the majority-held final-round answer across agents when the answer is automatically verifiable (numeric in GSM8k, multiple choice for the MMLU datasets) or the average final-round answer across agents otherwise (Biographies).

For GSM8k and the MMLU datasets, we report the proportion of samples in the dataset for which the generated answer exactly matches the ground-truth answer. For Biographies, following Du et al. (2023), we prompt an LLM judge (LLaMA-3.1-8B) to check whether each manually-decomposed fact in a ground-truth biography is supported (1), partially supported (0.5), or unsupported (0) in the generated biography, taking the mean of these scores over all facts as the per-biography accuracy and the mean over all dataset samples as the total accuracy.

**Comprehensive evaluation with the LLaMA family**   Table 3 presents results on each of the seven reasoning benchmarks across various baselines and activation communication. Notably, while NLD consistently outperforms LLaMA-3.2-3B, it does not always display a performance improvement over LLaMA-3.1-8B; but remarkably, AC *consistently*

---

[3]We use 3072 sentences as linear regression with $d$-dimensional input has a sample complexity of $O(d)$ (Vapnik, 1999).

*outperforms both single-model baselines*. In fact, AC offers an up to $27.0\%$ improvement *over NLD* across six of the seven reasoning datasets. When applying $W$ to $A$'s activation before performing the replacement function, we see even further gains of $2.6 - 50.0\%$ over vanilla AC for four of the seven datasets. We hypothesize that the benefits from the learned linear layer are less consistent across datasets because the subset of C4 data used to train $W$ likely contains text more semantically similar to some datasets than others, hence some datasets provide $W$ with out-of-distribution inputs which reduces performance compared to vanilla AC.

While we fix $A$ as the smaller model and $B$ as the larger model in Table 3 (so as to ensure decoding happens with the presumably more capable model), this need not be the case; swapping $A$ and $B$ yields results of $81.5 \pm 0.0$ and $61.0 \pm 4.8$ on Biographies and GSM8k respectively (without the linear layer). While these accuracies are lower than their non-swapped counterparts, notably they still are higher than both single-model baselines (and higher than NLD for Biographies); plus this is much more compute-efficient as the smaller model is now the one requiring the full instead of partial forward pass.

Note that we find **AC outperforms NLD on 48 of the 57 datasets in the full MMLU benchmark**; complete MMLU results, as well as a suite of additional experiments, are shown in Appendix B.

**Performance-compute tradeoff and generalization to different model scales**   Thus far, we have been considering the *absolute performance* of AC with respect to NLD, for which our method attains state-of-the-art results; however the superiority of activations as a language for inter-LLM

Table 3: **Accuracies (%) on all seven reasoning benchmarks.** NLD and all AC variants involve communication between LLaMA-3.2-3B ($A$) and LLaMA-3.1-8B ($B$); the performance of these models individually are presented in the first two rows of the table. NLD typically improves performance over at least one of the single model baselines; AC—*both* with and without the task-agnostic linear layer—consistently beats both baselines and NLD as well.

| Method | Biog. | GSM8k | HS Psych. | Logic | Col. Bio. | Prof. Law | Pub. Rel. |
|---|---|---|---|---|---|---|---|
| 3.2-3B | 79.4±0.0 | 58.0±4.9 | 30.0±1.0 | 16.0±0.8 | 11.0±0.7 | 0.0±0.0 | 26.0±0.1 |
| 3.1-8B | 83.9±0.0 | 60.0±4.9 | 65.0±0.1 | 42.0±0.1 | 50.0±0.2 | 20.0±0.8 | 53.0±0.2 |
| NLD | 80.2±0.1 | **75.0**±4.3 | 83.0±0.8 | 37.0±0.1 | 71.0±0.1 | 30.0±0.1 | 63.0±0.7 |
| AC | 84.6±0.0 | 64.0±4.8 | **85.0**±0.8 | **47.0**±0.1 | 78.0±0.9 | 30.0±0.1 | **74.0**±0.1 |
| AC ($\boldsymbol{W}$) | **86.8**±0.0 | 66.0±4.8 | 70.0±0.1 | 35.0±0.1 | **79.0**±0.9 | **45.0**±0.1 | 63.0±0.1 |

communication is further illustrated by AC's larger *ratio* of performance improvement to added inference-time compute over individual LMs. Figure 3 displays the results of single models, AC, and NLD across model scales and suites within the LLaMA family on the Biographies dataset. Incoming arrows to AC and NLD nodes denote the base models between which communication occurred. Not only does AC consistently outperform both single-model baselines unlike NLD, but also notice that the *slope* of each black line is far greater than the slope of each gray line, indicating that AC consistently achieves *greater increases in accuracy per additional unit of inference-time compute* (normalized by the compute of a single forward pass through LLaMA-3.2-1B on the given prompt) compared to NLD.

**Communication across model families**    Table 4 displays results for AC between models from the Qwen-2.5, Gemma-2, and LLaMA-3 families. We see that **AC beats NLD across the board**, and **beats both individual models for** $4/5$ **of the** $6$ **model pairs on Biographies/GSM8k respectively**—demonstrating the efficacy of AC irrespective of model architecture, size, tokenizer, and training data. Moreover, these results are obtained without training $\boldsymbol{W}$, meaning we do not need a separate projection layer between activation spaces to attain SOTA results, even for extremely distinct models! (We hypothesize this is because we are only replacing $B$'s last-token activation, hence $B$ can learn from $A$ without an extreme alteration to its activation distribution (Moschella et al., 2023; Kornblith et al., 2019).)

## 5. Conclusion

We present a simple approach to enable effective and computationally efficient communication between language models by injecting information from the activations of one model into the activations of another during the forward pass. Salient features of this approach include: (i) Scales up LLMs on new tasks by leveraging existing, frozen LLMs along with *zero* additional task-specific parameters and data, (ii) Applies to diverse domains and settings, and (iii) Saves a *substantial amount of compute*.

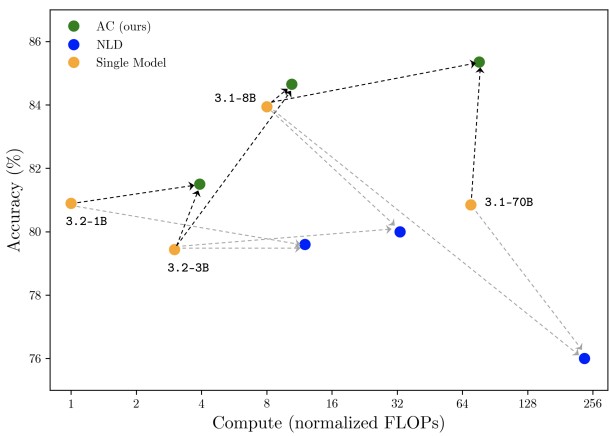

*Figure 3.* **Accuracy (%) vs. compute (# FLOPs normalized by single LLaMA-3.2-1B forward pass) for various configurations of AC and NLD on the Biographies dataset**. AC (🟢) yields the greatest performance gains per additional unit of inference-time compute over each baseline (🟡).

There are some limitations to this method. First, when not using the learned model-specific mapping discussed in Section 3.1, our method requires both models to have aligned embedding spaces, such that the activation of one model roughly retains its meaning in the other's activation space (note that unlike past works such as Pham et al. (2024) we do *not* require shared tokenizers or aligned vocabularies, only aligned embeddings). While less restrictive than past works (Pham et al., 2024), this assumption is somewhat limiting, but can be relaxed when we let $f$ be the learned model-specific mapping; and in practice we find that even amongst different models in the LLaMA family, no such mapping is required for state-of-the-art results.

Second, this method requires access to embeddings and will not work with black-box API access; however exploring API-only approaches is highly limiting, and recent releases of powerful open-source models (Dubey et al., 2024) merit the development of embedding-based techniques.

Table 4: **Individual model, AC, and NLD accuracies across three model families.** Each cell displays two values: Biographies score / GSM8k score.

| Model Pair $(A, B)$ | $A$ | $B$ | NLD | AC |
|---|---|---|---|---|
| LLaMA-3.2-3B, LLaMA-3.1-8B | 79.4±0.0 / 58.0±4.9 | 83.9±0.0 / 60.0±4.9 | 80.2±0.1 / **75.0**±4.3 | **84.6**±0.0 / 64.0±4.8 |
| Qwen-2.5-1.5B, Qwen-2.5-3B | 59.4±0.9 / 20.0±0.9 | 85.5±1.1 / 35.0±1.1 | 63.2±1.1 / 65.0±1.1 | **89.6**±1.0 / **70.0**±1.0 |
| Gemma-2-2B, Gemma-2-9B | 83.0±1.1 / 45.0±1.1 | **94.6**±0.9 / 80.0±0.9 | 70.3±1.0 / 70.0±1.0 | 88.1±0.7 / **90.0**±0.7 |
| Qwen-2.5-1.5B, LLaMA-3.2-3B | 59.4±0.9 / 20.0±0.9 | 79.4±0.0 / 58.0±4.9 | 75.4±1.0 / **75.0**±1.0 | **79.5**±1.0 / 75.0±1.0 |
| LLaMA-3.2-3B, Gemma-2-2B | 79.4±0.0 / 58.0±4.9 | 83.0±1.1 / 45.0±1.1 | 62.5±1.1 / 55.0±1.1 | **84.0**±0.1 / **60.0**±1.1 |
| Qwen-2.5-1.5B, Gemma-2-2B | 59.4±0.9 / 20.0±0.9 | **83.0**±1.1 / 45.0±1.1 | 49.3±1.1 / 50.0±1.1 | 73.0±1.1 / **55.0**±1.1 |

Third, while a concern might be the limited interpretability of communicating activations as opposed to natural language, we note the following. First, there is a fundamental tradeoff between interpretability and information preservation (as activations, by virtue of being much higher-dimensional than the space of natural language, allow proportionally higher-entropy communication) (Pham et al., 2024), which merits discussion beyond the scope of this work. But second, we actually posit that our method suggests a new avenue towards interpreting LM activations: "translating" activations based on the beliefs they induce as messages in listening agents, similar to the method put forward in Andreas et al. (2018). We recognize this as a promising avenue for future research.

Additional directions of future work include using AC to allow large LMs to leverage small, tunable LMs as "knowledge bases" during decoding (Lee et al., 2024), as in collaborative decoding (Shen et al., 2024) setups; and testing our approach on more complex coordination games (e.g., Lewis-style negotiation games (Lewis et al., 2017), Diplomacy).

## Impact Statement

This paper presents work whose goal is to advance the field of Machine Learning. There are many potential societal consequences of our work, none which we feel must be specifically highlighted here.

## Acknowledgements

The authors are grateful to Jacob Andreas, Yoon Kim, and Sham Kakade for their valuable discussions and feedback.

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

# A. Qualitative Results

> **Question**: *Give a bullet point biography of Joyce K. Reynolds, highlighting her contributions and achievements as a computer scientist.*

**Ground-Truth Biography:**
• *Joyce Kathleen Reynolds was an American computer scientist who contributed to the development of protocols underlying the Internet.*
• *She authored or co-authored many RFCs (Request for Comments) including Telnet, FTP, and POP protocols.*
• *Reynolds worked with Jon Postel to develop early functions of the Internet Assigned Numbers Authority and managed the root zone of DNS.*
• *She received the 2006 Postel Award for her services to the Internet.*
...

`LLaMA-3.2-3B`: • Joyce K. Reynolds is a renowned computer scientist and educator who has made significant contributions to the field of computer science and education.
 • She is a pioneer in the field of artificial intelligence, with a focus on human-computer interaction and user experience.
• Reynolds has written extensively on topics such as usability, accessibility, and human-centered design.
• She is a highly respected author, having written several books and articles on computer science and education.
• Reynolds has received numerous awards and honors for her contributions to the field.

`LLaMA-3.1-8B`: • Joyce K. Reynolds was born in 1923 in England. She studied mathematics at Girton College, Cambridge, graduating with a degree in 1945.
• Reynolds started her career in computing in 1945 at the National Physical Laboratory (NPL), where she worked on the development of the first commercial computer database, the Database Management System (DBMS).
• Reynolds was a key member of the team that developed the first DBMS, which was a pioneering achievement in the field of database management. Her work laid the foundation for modern database systems.

**AC**: • Joyce K. Reynolds is a renowned computer scientist and engineer, best known for her pioneering work in the development of the Internet's fundamental technologies.
• She was a key member of the team that designed the Internet Protocol (IP) and the Transmission Control Protocol (TCP), which form the basis of the modern Internet.
• Reynolds' work on IP and TCP enabled the creation of a network that could efficiently route and manage data packets, paving the way for the modern Internet as we know it.
• Reynolds has received numerous awards and honors for her groundbreaking work, including the Internet Society's (ISOC) highest award, the Jonathan B. Postel Service Award.
• She continues to be an influential figure in the field of computer science and engineering, inspiring new generations of scientists and engineers with her pioneering achievements.

*Figure 4.* **Example of AC on Biographies dataset.**

**Question**: *Christina is planning a birthday party and needs .75 gift bags per invited guest, because 1/4 of attendees don't show up. She invited 16 friends. Gift bags are $2 each. How much will she spend?*

`LLaMA-3.2-3B`: Since 1/4 of attendees don't show up, we need to calculate 3/4 of the total invited guests to find out how many will attend. 3/4 of 16 is (3/4) * 16 = 12.

Christina needs 0.75 gift bags per invited guest, so for 12 guests, she needs 0.75 * 12 = 9 gift bags.

Each gift bag costs $2, so the total cost will be 9 gift bags * $2 per bag = $18.

`LLaMA-3.1-8B`: Christina invited 16 friends, but 1/4 of them won't show up. To find out how many will attend, we need to calculate 1/4 of 16 and subtract that from 16.

To find 1/4 of 16, we divide 16 by 4. 16 / 4 = 4.

Now that we know 4 guests won't come, we subtract 4 from the total number of guests invited. 16 - 4 = 12 guests will attend.

Christina needs .75 gift bags per invited guest. Since she invited 16 friends, but only 12 will attend, we multiply .75 by 12. .75 * 12 = 9.

Gift bags are $2 each. Since Christina needs 9 gift bags, we multiply 9 by 2. 9 * 2 = $18.

**AC**: 1. Since 1/4 of the attendees don't show up, 3/4 of them will. To find the number of guests that will show up, we need to multiply the total number of invited guests by 3/4. So, 16 * 3/4 = 12 guests will show up.
2. Christina needs 0.75 gift bags per invited guest. Since she invited 16 friends, she needs 16 * 0.75 = 12 gift bags.
3. Each gift bag costs $2. Since she needs 12 gift bags, the total cost will be 12 * $2 = $24.

*Figure 5.* **Example of AC on GSM8k dataset.**

**Question**: *According to gate-control theory, which condition tends to close the gate?*
A) anxiety, B) depression, C) focusing on the pain, D) electrical stimulation

`LLaMA-3.2-3B`: The question is about the gate-control theory of pain, which suggests that certain stimuli can close the gate to the central nervous system, thereby reducing the perception of pain. The options are all conditions or stimuli that can affect the perception of pain. The correct answer is the one that directly affects the gate, which is the focus on the pain. When we focus on the pain, it tends to close the gate, thereby reducing the perception of pain. The other options are not directly related to the gate in the same way. Anxiety and depression can affect pain perception, but they do not directly close the gate. Electrical stimulation can actually activate the gate, not close it. Therefore, the correct answer is C) focusing on the pain.

`LLaMA-3.1-8B`: Gate-control theory suggests that the gate (the gate in the spinal cord that regulates pain) can be closed by the activation of certain nerve fibers that carry nonpainful sensations. Anxiety, depression, and focusing on the pain can all increase pain perception, which would suggest that the gate is open, not closed. Electrical stimulation, on the other hand, can activate certain nerve fibers that carry nonpainful sensations, which can close the gate and reduce pain perception. Therefore, the correct answer is D) electrical stimulation.

**AC**: Step 1: Understand the gate-control theory. The gate-control theory of pain proposes that the nervous system has a "gate" that controls the flow of pain signals to the brain. The theory suggests that this gate can be opened or closed by various factors. Step 2: Identify the conditions that tend to close the gate. According to the gate-control theory, conditions that tend to close the gate include factors that reduce the transmission of pain signals to the brain. These factors can include distraction, relaxation, and certain types of stimulation. Step 3: Analyze the options. Anxiety (A) tends to increase the transmission of pain signals to the brain, making it more likely to open the gate. Depression (B) can have a complex relationship with pain perception, but it is not directly related to closing the gate. Focusing on the pain (C) tends to increase the transmission of pain signals to the brain, making it more likely to open the gate. Electrical stimulation (D) can be used to close the gate by overriding the pain signal with a more intense stimulus. The final answer is: <<D>>

*Figure 6.* **Example of AC on MMLU High School Psychology dataset.**

**Question**: *Identify the conclusion of the following argument. From a historical point of view, genetics is a young science. The principles that govern inheritance traits were described less than 150 years ago. Furthermore, around the turn of the twentieth century, the laws of inheritance were rediscovered. But even then, the importance of DNA was not really understood until the 1950s.*
*A) But even then, the importance of DNA was not really understood until the 1950s., B) The importance of DNA was not really understood until the 1950s., C) From a historical point of view, genetics is a young science., D) The principles that govern inheritance traits were described less than 150 years ago.*

`LLaMA-3.2-3B`: The best answer is B) The importance of DNA was not really understood until the 1950s.

`LLaMA-3.1-8B`: The final answer is: <<A>>. Explanation: The conclusion is the statement that follows from the premises. In this case, the conclusion is the statement that follows from the two premises. The first premise states that genetics is a young science from a historical point of view, and the second premise states that the importance of DNA was not really understood until the 1950s. The conclusion is the statement that summarizes the main point of the argument, which is that even then, the importance of DNA was not really understood until the 1950s. This is the correct answer because it is the statement that follows from the premises and summarizes the main point of the argument.

**AC**: Step 1: Identify the main claim of the argument. The main claim of the argument is that genetics is a young science from a historical point of view. Step 2: Identify the supporting evidence for the main claim.The supporting evidence includes the fact that the principles that govern inheritance traits were described less than 150 years ago and that the laws of inheritance were rediscovered around the turn of the twentieth century. Step 3: Identify the additional information provided in the argument. The additional information is that the importance of DNA was not really understood until the 1950s. Step 4: Determine the conclusion of the argument. The conclusion is the main claim of the argument, which is that genetics is a young science from a historical point of view. <<C>>

*Figure 7.* **Example of AC on MMLU Formal Logic dataset.**

**Question**: *The members of a bird species capable of nesting in treetops or on the ledges of nearby cliffs make their homes only in the treetops due to the established presence of a more aggressive bird species on the cliff ledges. With respect to habitat, the treetop-nesting birds are utilizing: A) their fundamental niche only, B) their realistic niche only, C) their ecological niche only, D) neither their fundamental niche nor their realistic niche.*

`LLaMA-3.2-3B`: The treetop-nesting birds are not utilizing their fundamental niche because the cliff ledge is not part of their natural environment. However, they are utilizing their realistic niche, which is the portion of the fundamental niche that is actually available to the birds. The treetop-nesting birds are not utilizing their ecological niche, which is the actual area used by the birds. Therefore, the correct answer is B) their realistic niche only.

`LLaMA-3.1-8B`: The birds are using their ecological niche, which is the set of environmental conditions and resources that a species can use and occupy in a given area. The birds are not using their fundamental niche, which is the set of all environmental conditions and resources that a species can use and occupy in theory. The birds are also not using their realistic niche, which is the set of environmental conditions and resources that a species can use and occupy in a given area, but with the constraint of competition and predation. The birds are only using their ecological niche because they are limited by the presence of a more aggressive bird species on the cliff ledges. Therefore, the correct answer is C) their ecological niche only.

**AC**: Step 1: Define the key terms: fundamental niche, realistic niche, and ecological niche. The fundamental niche refers to the range of environmental conditions in which a species can survive and reproduce without any limitations. The realistic niche, on the other hand, is the range of conditions that a species actually occupies due to biotic and abiotic interactions with other species. The ecological niche is a broader concept that encompasses both the fundamental and realistic niches. Step 2: Analyze the given scenario. In the given scenario, the bird species capable of nesting in treetops or on cliff ledges makes their homes only in the treetops due to the presence of a more aggressive bird species on the cliff ledges. This means that the treetop-nesting birds are limited by the presence of another species in their potential habitat. Step 3: Determine the type of niche being utilized. Since the treetop-nesting birds are limited by the presence of another species in their potential habitat, they are not utilizing their fundamental niche or ecological niche, but are utilizing B) only their realistic niche, which is the range of conditions that they actually occupy due to biotic and abiotic interactions with other species.

*Figure 8.* **Example of AC on MMLU College Biology dataset.**

**Question**: *A purchaser orally agreed with the seller to buy seller's house for $100,000. The purchaser gave the seller a written offer and a post-dated check for 5% of the price. Before the seller signed the offer form or deposited the check, he received an offer of his full asking price of $120,000 from another party. The seller called the purchaser and called the deal off. The purchaser sued for specific performance on the basis that the oral understanding, along with the written offer and the check, constituted an enforceable agreement. Will the purchaser likely prevail?* A) Yes, because there was an oral agreement and the check for the down payment was evidence of consideration being paid, B) No, because the offer was not accepted—prior to an acceptance the seller had the right to sell to anyone, C) Yes, because the buyer and seller had a meeting of the minds evidenced by their oral understanding, the check and the written offer, D) No, because specific performance does not apply in a residential real estate transaction.

`LLaMA-3.2-3B`: The most accurate answer would be: A). Here's the explanation: The purchaser and seller had an oral agreement to buy the house for $100,000, which is a sufficient consideration to support the enforceability of the agreement. The written offer and the post-dated check for 5% of the price ($5,000) also constitute consideration, as they demonstrate the purchaser's commitment to the deal. The seller's receipt of the written offer and the post-dated check prior to the offer from the other party indicates that the seller had already accepted the purchaser's offer and was willing to move forward with the deal. Therefore, the purchaser can argue that the seller had already accepted the offer and was not free to accept the other offer.

`LLaMA-3.1-8B`: The purchaser and seller had a valid contract. The check was consideration, and the oral agreement was enforceable. The seller had no right to call off the deal after the purchaser's offer was made. The seller could have sold to anyone, but he was not free to call off the deal with the purchaser after the purchaser's offer was made. The purchaser had a right to enforce the contract. <<A>>

**AC**: The purchaser will not prevail because the offer was not accepted. The seller received another offer and called off the deal before accepting the purchaser's offer. The law requires that there be an acceptance of the offer before a contract can be formed. The seller's action in calling off the deal before accepting the purchaser's offer terminated the offer, and the purchaser's offer lapsed. <>

*Figure 9.* **Example of AC on MMLU Professional Law dataset.**

**Question**: *No-Wait Weight Loss, a diet pill brand, promises its potential consumers that if they take one pill once a day they will lose weight without having to change their diet or start exercising. However, many people who have used the product have complained that they haven't lost weight, and some have even gained weight. No-Wait Weight Loss has been unsuccessful in the _________ stage of the advertising campaign: A) distribution, B) penetration, C) participation, D) identification*

`LLaMA-3.2-3B`: The best answer is B). Explanation: The product has failed to gain traction with consumers, which is referred to as the penetration stage of the advertising campaign. The penetration stage is the point in time when a product is introduced to the market and has not yet reached a significant level of adoption or usage. In this case, the product is not gaining any traction, and many consumers are not using it, which indicates that it has not yet reached the penetration stage of the advertising campaign.

`LLaMA-3.1-8B`: We need to consider the stages of the advertising campaign and how they relate to the product's performance. The stages are: 1) Identification: The initial stage... 7) Participation: The stage where consumers become engaged with the brand. We can infer that the product has not been able to convince potential consumers of its benefits, which is the Persuasion stage. However, this is not the correct answer. The product has been used by many people, and some have even complained about its performance, which suggests that the product has reached the Trial stage. However, this is also not the correct answer. The product has been unsuccessful in the stage of the advertising campaign, which suggests that the product has not been able to gain market share and become a dominant player in the market, which is the Penetration stage. Therefore, the correct answer is B) Penetration.

**AC**: The question is asking about the stage of the advertising campaign that No-Wait Weight Loss has been unsuccessful in. The stages of the advertising campaign are: identification, awareness, knowledge, liking, persuasion, trial, and adoption. However, the options provided in the question are not the traditional stages of the advertising campaign. The options are distribution, penetration, participation, and identification. However, if we map the options to the traditional stages of the advertising campaign, we can see that: distribution is similar to the trial stage, penetration is similar to the adoption stage, participation is similar to the awareness stage, and identification is similar to the awareness stage as well. Since No-Wait Weight Loss has been unsuccessful in getting people to buy and use their product, it has been unsuccessful in the trial stage of the advertising campaign, which is similar to the distribution stage. Therefore, the correct answer is <<A>>.

*Figure 10.* **Example of AC on MMLU Public Relations dataset.**

Table 5: **Reasoning benchmark performance when varying tokens modified during AC.** All methods involve communication between LLaMA-3.2-3B ($A$) and LLaMA-3.1-8B ($B$). The functional form $f$ is varied between last-token replacement, last-token summation, and summation for all tokens.

| Method | Biog. | GSM8k | HS Psych. | Logic | Col. Bio. | Prof. Law | Pub. Rel. |
|---|---|---|---|---|---|---|---|
| **AC (replace)** | **84.6**±0.0 | 64.0±4.8 | **85.0**±0.8 | **47.0**±0.1 | **78.0**±0.9 | **30.0**±0.1 | **74.0**±0.1 |
| **AC (sum)** | 79.7±0.0 | **66.0**±4.7 | 65.0±4.8 | 42.0±4.9 | 50.0±5.0 | 25.0±4.3 | 37.0±4.8 |
| **AC (all tokens)** | 76.0±0.0 | 62.0±4.9 | 35.0±4.8 | 42.0±4.9 | 61.0±4.9 | 15.0±3.6 | 26.0±4.4 |

Table 6: **Reasoning benchmark performance when sampling from $A$ with CoT.** All methods involve communication between LLaMA-3.2-3B ($A$) and LLaMA-3.1-8B ($B$).

| Method | Biog. | GSM8k | HS Psych. | Logic | Col. Bio. | Prof. Law | Pub. Rel. |
|---|---|---|---|---|---|---|---|
| **AC** | 84.6±0.0 | 64.0±4.8 | **85.0**±0.8 | **47.0**±0.1 | 78.0±0.9 | 30.0±0.1 | **74.0**±0.1 |
| **AC ($W$)** | **86.8**±0.0 | **66.0**±4.8 | 70.0±0.1 | 35.0±0.1 | **79.0**±0.9 | **45.0**±0.1 | 63.0±0.1 |
| **AC (CoT)** | 82.1±0.0 | **66.0**±4.0 | 80.0±4.0 | 26.0±4.4 | 67.0±4.7 | 40.0±4.9 | 63.0±4.8 |

# B. Additional Experiments

## B.1. Modifying Activations of All Tokens

Recall that AC grafts the last-token layer-$k$ activation of $A$ into $B$'s last-token layer-$j$ activation. But is modifying just the last token activation enough to communicate information from $A$ to $B$?

Note that after applying masked attention in each of the previous Transformer layers, the last token activation of $A$ attends to all tokens before it, hence incorporating information from the entire sequence. Indeed, this must be the case for activation communication to recover the gap between the zero-communication and skyline setups on both coordination games, which (for Tip Sheets in particular) require information starting at the first few tokens of $A$'s prompt to be communicated.

To verify this empirically, we experiment with summing the activations of all tokens in the sequence rather than just the last (we cannot replace all tokens as this would just replace $B$'s layer-$j$ activation with $A$'s layer $k$-activation). Results are shown in Table 5.

Indeed, applying $f$ to all tokens **decreases** performance relative to applying $f$ to just the last token. Note that the fact performance generally decreases from $f = \texttt{replace}$ to $f = \texttt{sum}$, and further with all tokens, is expected. The high performance of AC with $f = \texttt{replace}$ means that the edited last-token activation $\boldsymbol{b}$ retains some meaning in $B$'s activation space; it is less likely for this to be the case when $f = \texttt{sum}$ (at the very least $\boldsymbol{b}$ has norm roughly 2× that of $B$'s original last-token activation), and when doing this for all tokens we'd expect performance to decrease even further as now all activation vectors, not just the last, are out-of-distribution with respect to $B$'s activation space.

## B.2. Incorporating Chain-of-Thought Prompting

How does AC perform in relation to NLD in cases where $A$ might incur a long response (possibly with chain-of-thought for intermediate answer computation)? I.e., does AC lose out on the benefits of CoT?

First, note that we still reap the benefits of CoT when we sample a completion from $B$ after AC (where $B$ gets all the information encoding $A$'s "beliefs" about the prompt via AC, hence CoT on $A$'s side is not needed). To verify this, we experiment with prompting $A$ with CoT, generating a full response, and then passing the layer-$k$ last-token activation of the *CoT response* to $B$ as part of AC. Results are shown in Table 6.

Indeed, we empirically find our above intuition (in orange) to hold, as there is no significant improvement over vanilla AC when generating from $A$ using CoT.

## B.3. Learning $W$ In-Distribution

Recall our reasoning about the AC ($W$) results from Section 4.2: "We hypothesize that the benefits from the learned linear layer are less consistent across datasets because the subset of C4 data used to train $W$ likely contains text more

Table 7: **GSM8k performance when learning $W$ in-distribution.** All AC variants involve communication between `LLaMA-3.2-3B` ($A$) and `LLaMA-3.1-8B` ($B$).

| AC | AC ($W$) | AC ($W_{\text{in dist}}$) |
|---|---|---|
| $64.0\pm4.8$ | $66.0\pm4.8$ | $\mathbf{78.0}\pm4.1$ |

Table 8: **Reasoning benchmark performance of communication between identical models.** Both NLD and AC involve communication between 2 instances of `LLaMA-3.1-8B`. 512-token completions are sampled with temperature 0.7 and debate is run for 2 rounds.

| Method | Biog. | GSM8k | HS Psych. | Logic | Col. Bio. | Prof. Law | Pub. Rel. |
|---|---|---|---|---|---|---|---|
| `LLaMA-3.1-8B` | $\mathbf{83.9}\pm0.0$ | $60.0\pm4.9$ | $65.0\pm0.1$ | $\mathbf{42.0}\pm0.1$ | $50.0\pm0.2$ | $20.0\pm0.8$ | $53.0\pm0.2$ |
| **NLD** | $80.8\pm0.0$ | $\mathbf{70.0}\pm3.7$ | $\mathbf{85.0}\pm3.6$ | $35.0\pm4.8$ | $\mathbf{78.0}\pm4.1$ | $\mathbf{40.0}\pm4.9$ | $53.0\pm5.1$ |
| **AC** | $83.7\pm0.0$ | $60.0\pm4.9$ | $\mathbf{85.0}\pm3.6$ | $40.0\pm4.9$ | $74.0\pm4.4$ | $\mathbf{40.0}\pm4.9$ | $\mathbf{79.0}\pm4.1$ |

semantically similar to some datasets than others, hence some datasets provide $W$ with out-of-distribution inputs which reduces performance compared to vanilla AC."

Indeed, we verify this hypothesis by training $W$ on the GSM8k train set (to produce $W_{\text{in dist}}$) and then evaluating with this task-specific linear layer on the GSM8k test set. Results are shown in Table 7.

Indeed, learning $W$ in-distribution significantly boosts performance, confirming our hypothesis. Unfortunately we cannot run this experiment for the other datasets, as there is no in-distribution training data available for MMLU (we use all public data for testing).

Hence, this suggests that AC ($W$) should unilaterally improve over vanilla AC if we choose a training set with good coverage across many tasks and distributions, such that there are sentences semantically similar to prompts across the span of downstream task datasets.

### B.4. Activation Space Similarity $\propto$ AC Performance Gain

We conduct the following experiment: for each of the six pairs of models $A$, $B$ in the above experiment (see Table 4), we compute the increase in Biographies performance with AC relative to the average individual performance of $A$ and $B$. We also compute the matrix analogue of the squared cosine similarity between the models' activation spaces,

$$\frac{\|Y^\top X\|_F^2}{\|X\|_F^2 \, \|Y\|_F^2},$$

where $X$ is the matrix of $A$'s activations on 3072 sentences from C4 (the same dataset used to train $W$), $Y$ is the corresponding matrix for $B$, and $\|\cdot\|_F$ denotes the Frobenius norm. This yields the plot in Figure 11.

There is a clear positive correlation between the similarity of the activation distributions and the AC performance gain, as expected; the more aligned $A$ and $B$'s activation spaces are, the more semantically meaningful and useful the embedding we graft from $A$ to $B$ becomes.

### B.5. Communicating Activations Between Identical Models

Note that AC as described in Section 3.1 only supports communication between distinct models. We can extend AC to work for communication between identical models as follows: let $A$ and $B$ be instances of the same model. We can sample a completion from $A$ with temperature and graft the last-token layer-$k$ activation of the *completion* into $B$ at layer $j$ as part of the AC procedure. This still saves a substantial amount of compute over NLD between 2 model instances, showing our technique can apply to this setting. Table 8 shows the results of this experiment.

Indeed, while communication between multiple model instances doesn't always show improvement over the single model itself (a well-known result from (Du et al., 2023)), **AC matches/outperforms NLD on five of the seven datasets**.

The intuition behind debate between multiple identical model instances is that sampling multiple completions (with temperature) from the same model yields diverse reasoning paths that can be recombined into a stronger final answer. The

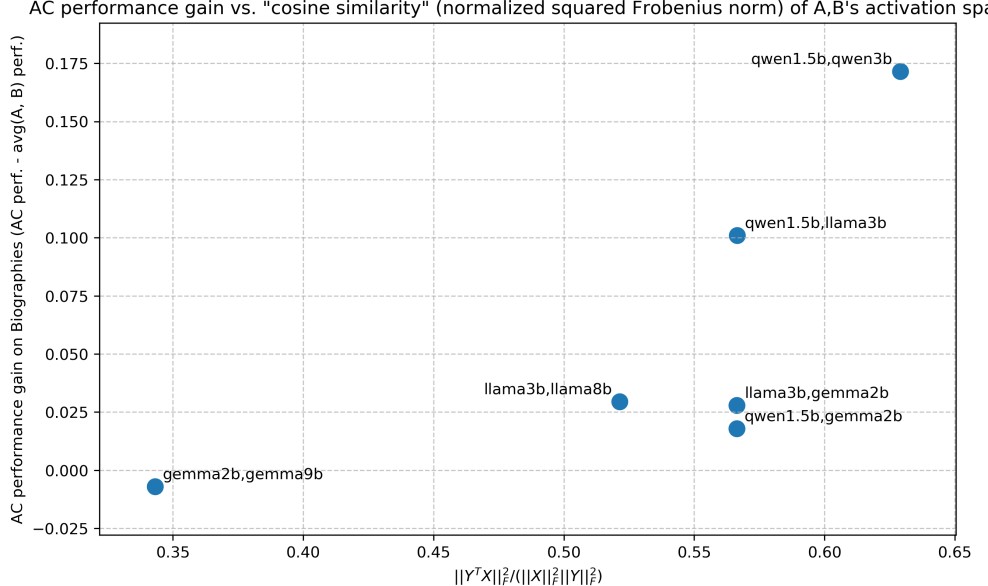

*Figure 11.* AC performance gain over average $A/B$ individual performance on Biographies, as a function of matrix "cosine similarity" between $A$ and $B$'s activation spaces.

Table 9: **Reasoning benchmark performance of AC and NLD with varying number of rounds.** All methods involve communication between LLaMA-3.2-3B ($A$) and LLaMA-3.1-8B ($B$).

| Method | Biog. | GSM8k | HS Psych. | Logic | Col. Bio. | Prof. Law | Pub. Rel. |
|---|---|---|---|---|---|---|---|
| **NLD (1 round)** | 83.6±0.0 | 72.0±4.5 | 65.0±4.8 | 40.0±4.9 | 68.0±4.6 | 30.0±4.6 | 63.0±4.8 |
| **NLD (2 rounds)** | 80.2±0.1 | 75.0±4.3 | 83.0±0.8 | 37.0±0.1 | 71.0±0.1 | 30.0±0.1 | 63.0±0.7 |
| **NLD (3 rounds)** | 80.1±4.6 | **79.0**±4.1 | 70.0±4.6 | 45.0±5.0 | 63.0±4.8 | **40.0**±4.9 | **74.0**±4.4 |
| **NLD (4 rounds)** | 78.0±0.0 | **79.0**±4.1 | * | * | * | * | * |
| **AC** | **84.6**±0.0 | 64.0±4.8 | **85.0**±0.8 | **47.0**±0.1 | **78.0**±0.9 | 30.0±0.1 | **74.0**±0.1 |

*Runs required too much compute

above experiment shows that the same intuition holds for AC—we are sampling multiple times from the same model, but passing responses between agents via AC rather than as NL messages.

### B.6. Additional Rounds of Natural Language Debate

In Section 4.2 we fix NLD to 2 agents and 2 rounds, however we find in additional experiments that AC outperforms NLD even with additional rounds, highlighting the superiority and robustness of activations as an alternative "language" for inter-LM communication. Results are shown in Table 9; we see that for 5 of the 7 reasoning benchmarks, AC beats NLD even with 3-4 rounds while using *substantially* less compute.

### B.7. Full MMLU Benchmark Results

Table 10 below displays complete results of both AC and NLD on the full MMLU benchmark. Notably, **AC matches/outperforms NLD on 48/57 datasets, with substantially less compute used**, indicating its superiority and robustness as an alternative "language" for inter-LLM communication.

Table 10: **Comparison of NLD vs. AC on the full MMLU benchmark (Hendrycks et al., 2021).**

| Dataset | NLD | AC |
|---|---|---|
| Conceptual Physics | $60.0 \pm 4.9$ | $\mathbf{68.0 \pm 4.6}$ |
| High School Chemistry | $\mathbf{50.0 \pm 5.0}$ | $37.0 \pm 4.8$ |
| Security Studies | $60.0 \pm 4.9$ | $60.0 \pm 4.9$ |
| Jurisprudence | $84.0 \pm 3.6$ | $84.0 \pm 3.6$ |
| Logical Fallacies | $63.0 \pm 4.8$ | $\mathbf{72.0 \pm 4.5}$ |
| College Computer Science | $44.0 \pm 5.0$ | $44.0 \pm 5.0$ |
| International Law | $55.0 \pm 5.0$ | $\mathbf{59.0 \pm 4.9}$ |
| Miscellaneous | $90.0 \pm 3.0$ | $\mathbf{95.0 \pm 2.2}$ |
| Marketing | $70.0 \pm 4.6$ | $\mathbf{85.0 \pm 3.6}$ |
| Elementary Mathematics | $\mathbf{75.0 \pm 4.3}$ | $58.0 \pm 4.9$ |
| Machine Learning | $42.0 \pm 4.9$ | $42.0 \pm 4.9$ |
| High School Macroeconomics | $44.0 \pm 5.0$ | $\mathbf{75.0 \pm 4.3}$ |
| High School US History | $45.0 \pm 5.0$ | $\mathbf{71.0 \pm 4.6}$ |
| Human Aging | $56.0 \pm 5.0$ | $\mathbf{72.0 \pm 4.5}$ |
| Astronomy | $79.0 \pm 4.1$ | $\mathbf{80.0 \pm 4.0}$ |
| Computer Security | $56.0 \pm 5.0$ | $\mathbf{75.0 \pm 4.3}$ |
| High School Statistics | $\mathbf{55.0 \pm 5.0}$ | $42.0 \pm 4.9$ |
| Professional Medicine | $\mathbf{79.0 \pm 4.1}$ | $65.0 \pm 4.8$ |
| Electrical Engineering | $58.0 \pm 4.9$ | $\mathbf{60.0 \pm 4.9}$ |
| High School Computer Science | $63.0 \pm 4.8$ | $\mathbf{70.0 \pm 4.6}$ |
| College Physics | $\mathbf{50.0 \pm 5.0}$ | $28.0 \pm 4.5$ |
| Management | $74.0 \pm 4.1$ | $\mathbf{75.0 \pm 4.3}$ |
| Moral Scenarios | $40.0 \pm 4.9$ | $40.0 \pm 4.9$ |
| World Religions | $58.0 \pm 4.9$ | $\mathbf{72.0 \pm 4.5}$ |
| Virology | $47.0 \pm 5.0$ | $\mathbf{50.0 \pm 5.0}$ |
| Philosophy | $67.0 \pm 4.7$ | $\mathbf{70.0 \pm 4.6}$ |
| Abstract Algebra | $\mathbf{50.0 \pm 5.0}$ | $28.0 \pm 4.5$ |
| High School Government and Politics | $\mathbf{80.0 \pm 4.0}$ | $61.0 \pm 4.9$ |
| High School Biology | $60.0 \pm 4.9$ | $\mathbf{65.0 \pm 4.8}$ |
| College Mathematics | $64.0 \pm 4.8$ | $\mathbf{66.0 \pm 2.4}$ |
| Global Facts | $33.0 \pm 5.0$ | $\mathbf{37.0 \pm 4.8}$ |
| High School World History | $71.0 \pm 4.0$ | $\mathbf{74.0 \pm 4.4}$ |
| High School European History | $68.0 \pm 4.0$ | $\mathbf{71.0 \pm 4.6}$ |
| College Medicine | $\mathbf{65.0 \pm 4.8}$ | $53.0 \pm 5.0$ |
| High School Geography | $67.0 \pm 4.7$ | $\mathbf{79.0 \pm 4.1}$ |
| Anatomy | $74.0 \pm 4.4$ | $74.0 \pm 4.4$ |
| Human Sexuality | $75.0 \pm 4.3$ | $75.0 \pm 4.3$ |
| Medical Genetics | $79.0 \pm 4.1$ | $\mathbf{82.0 \pm 3.8}$ |
| Professional Accounting | $40.0 \pm 4.9$ | $\mathbf{48.0 \pm 4.5}$ |
| US Foreign Policy | $89.0 \pm 3.1$ | $\mathbf{90.0 \pm 3.1}$ |
| Business Ethics | $43.0 \pm 5.0$ | $\mathbf{44.0 \pm 5.0}$ |
| College Chemistry | $41.0 \pm 5.0$ | $\mathbf{47.0 \pm 5.0}$ |
| High School Physics | $40.0 \pm 5.0$ | $\mathbf{47.0 \pm 5.0}$ |
| Professional Psychology | $54.0 \pm 4.8$ | $\mathbf{55.0 \pm 5.0}$ |
| Sociology | $68.0 \pm 4.1$ | $\mathbf{68.0 \pm 4.6}$ |
| High School Microeconomics | $95.0 \pm 2.2$ | $95.0 \pm 2.2$ |
| High School Mathematics | $55.0 \pm 5.0$ | $55.0 \pm 5.0$ |
| Prehistory | $\mathbf{75.0 \pm 4.3}$ | $60.0 \pm 4.9$ |
| Nutrition | $64.0 \pm 4.5$ | $\mathbf{70.0 \pm 4.6}$ |
| Clinical Knowledge | $65.0 \pm 4.3$ | $65.0 \pm 4.8$ |
| Moral Disputes | $58.0 \pm 4.8$ | $\mathbf{60.0 \pm 4.9}$ |
| Econometrics | $40.0 \pm 5.0$ | $40.0 \pm 4.9$ |
| High School Psychology | $83.0 \pm 0.8$ | $\mathbf{85.0 \pm 0.8}$ |
| Formal Logic | $37.0 \pm 0.1$ | $\mathbf{47.0 \pm 0.1}$ |
| College Biology | $71.0 \pm 0.1$ | $\mathbf{78.0 \pm 0.9}$ |
| Professional Law | $30.0 \pm 0.1$ | $30.0 \pm 0.1$ |
| Public Relations | $63.0 \pm 0.7$ | $\mathbf{74.0 \pm 0.1}$ |
| **Average** | $60.7 \pm 2.0$ | $\mathbf{62.7 \pm 2.2}$ |

