# OpenReview forum: "Communicating Activations Between Language Model Agents"
_ICML.cc/2025/Conference — ICML 2025 poster_

### Official Review · Reviewer_9iZt · 2025-03-02

**Overall Recommendation:** 3

**Summary:**

This paper studies the multi-agent communication problem in the LLM scenario. Specifically, it proposes using hidden representations instead of natural language. Experiments on several multi-agent collaboration datasets demonstrate the effectiveness of the proposed method.

**Claims And Evidence:**

Yes

**Essential References Not Discussed:**

N/A

**Experimental Designs Or Analyses:**

Yes. The experiments on the multi-agent database scenario make sense and can be used to demonstrate the effectiveness of the proposed method.

**Methods And Evaluation Criteria:**

Yes

**Other Comments Or Suggestions:**

N/A

**Other Strengths And Weaknesses:**

Strength:
The paper is well-motivated and clearly written.

Limitation:
1. The theoretical depth of this paper can be further enhanced. For example, the result shows that the direct replacement strategy is the most efficient way. Basically, it means that we should discard all previous representations and directly use the projected ones. Do we have any assumptions behind this phenomenon? (e.g., what is the relationship between the two models such that this conclusion holds?)
2. Many design choices need further clarification. For details, please see the question for authors section.

**Questions For Authors:**

1. What is the connection and difference of your model against the encoder-decoder model?
2. How to choose the hyperparameters of j,k
3. What is the effect of the scale and distribution of the aligning data for training the transformation matrix?
4. Are you trying to train a unique matrix for each j,k pair? Since you are using the MSE loss, I wonder if such a training objective might work well given that the capacity of a matrix is not very big.
5. Have you tried directly aligning the representations from two models and adding one more layer on top of that?

**Relation To Broader Scientific Literature:**

The multi-agent communication problem has been a long-term topic in the AI community. Recently, due to the development of LLMs, people have started using natural language as a communication medium. As discussed in the paper, previous work already discussed that the natural language might not be the most efficient way and this paper follows the trend to use hidden representation but with a more careful design (e.g., train a transformation matrix)

**Theoretical Claims:**

N/A no theoretical claims

---

> ### Author Rebuttal · Authors · 2025-04-01
>
> We thank the reviewer for their feedback, and address their comments below:
>
> 1. > The theoretical depth of this paper can be further enhanced. For example, the result shows that the direct replacement strategy is the most efficient way. Basically, it means that we should discard all previous representations and directly use the projected ones. Do we have any assumptions behind this phenomenon? (e.g., what is the relationship between the two models such that this conclusion holds?)
>
> First, note that the direct replacement strategy only replaces $B$'s layer-$j$ activation of *the final token* with $A$'s last-token layer-$k$ activation. The embeddings at all other token positions remain the same. Hence, we are not actually discarding all previous representations: after applying masked attention in each of the previous Transformer layers, the last token activation of $A$ attends to all tokens before it, hence incorporating information from the entire sequence; and the previous token activations of $B$ are retained and this incorporate all of $B$'s "thoughts" regarding the full sequence.
>
> Please refer to Appendix B.1 for additional discussion and comparison between the replacement, sum, and mean strategies for activation grafting.
>
> 2. > What is the connection and difference of your model against the encoder-decoder model?
>
> This is an interesting point; the direct replacement activation grafting approach could be seen as running an encoder-decoder Transformer, where the first $k$ layers of $A$ are the encoder, $B$ is the decoder, and we have a "special cross-attention" where----from layer $j$ onwards----all tokens in $B$ attend to only the last-token embedding outputted by the first $k$ layers of $A$. However $A$ and $B$ are language models used out-of-the-box, with no joint training, no requirement of shared tokenizer/aligned embedding space/similar training distributions/etc.
>
> 3. > How to choose the hyperparameters of j,k
>
> **Please see point #3 in the reply to Reviewer VKxi above.**
>
> 4. > What is the effect of the scale and distribution of the aligning data for training the transformation matrix?
>
> We propose training $\mathbf{W}$ to minimize MSE loss over a dataset of $N$ sentences
> $\frac{1}{N}\sum_{i=1}^N \left\|\mathbf{z}^{(i)} - \mathbf{W}\mathbf{y}^{(i)}\right\|_2^2$ where each $(\mathbf{y}^{(i)},\mathbf{z}^{(i)})$ pair denotes the final-token layer-$26$ activations of $A$ and $B$ at layers $k$ and $j$ respectively given the same sentence as input.
>
> Hence, the "aligning data distributions" are quite literally $A$ and $B$'s activation distributions; we do not scale or otherwise modify the activation vectors in any way before training.
>
> 5. > Are you trying to train a unique matrix for each j,k pair? Since you are using the MSE loss, I wonder if such a training objective might work well given that the capacity of a matrix is not very big.
>
> Yes we are, see point #4 above. However, note that for each model pair, we only need a single $(j,k)$ pair----indeed, we attain SOTA results with $k=j=26$ fixed.
>
> Also, this training objective is quite standard in related literature. For instance, the "model stitching" paper [1] learns a 1x1 convolution between activations of two models at a specific layer, which is exactly equivalent to our method of training $\mathbf{W}$; [2] also learns a linear projection between Transformer layers which is shown to be quite expressive.
>
> Indeed, we find our approach yields quite strong results; please see Appendix B.3 for additional discussion.
>
> 6. > Have you tried directly aligning the representations from two models and adding one more layer on top of that?
>
> We kindly ask for clarification on this question. If by "directly aligning the representations from two models and adding one more layer on top of that" you mean learning a linear layer that projects activations from $A$'s activation space to $B$'s, that is exactly what we do in the AC ($\mathbf{W}$) method.
>
> **We hope that with the provided responses & additional strong results, you consider raising your score to support clear acceptance. Please let us know if there are any additional questions or concerns; we'd be happy to address them.**
>
> [1] Yamini Bansal, Preetum Nakkiran, Boaz Barak. Revisiting Model Stitching to Compare Neural Representations, 2021.
>
> [2] Alexander Yom Din et al. Jump to Conclusions: Short-Cutting Transformers With Linear Transformations, 2024.

---

### Official Review · Reviewer_EeXt · 2025-03-11

**Overall Recommendation:** 4

**Summary:**

The paper proposes a novel method for inter-language model (LM) communication by directly exchanging model activations instead of using natural language.
Tested on two synthetic datasets (a coordination game and an investment decision task) and several reasoning benchmarks (GSM8k, MMLU subsets, Biographies), the method achieves up to 27% improvement over natural language communication while using less than one-fourth the compute. The paper compares activation communication with natural language debate, a common multi-agent method, showing superior efficiency and generalization across model sizes.

## update after rebuttal

I have considered the additional set of experiments and the preliminary analysis and I don't see any reasons why this paper shouldn't be accepted. I am increasing my score from 3 to 4

**Claims And Evidence:**

The claims in the paper are partially supported by the experiments, which demonstrate that activation communication improves performance while reducing compute costs compared to natural language communication. The results across two synthetic datasets and multiple reasoning benchmarks provide some evidence for the claimed efficiency and effectiveness of the approach. The ablation studies on different activation combination functions further strengthen the argument that direct activation exchange can enhance inter-model communication. The application to Llama-only models make it for a partial support of the general claim that their methods can improve LLM performance

**Essential References Not Discussed:**

Most essential references are properly discussed.
A link that I didn't see in the paper is the one to papers like Eliciting Latent Predictions from Transformers with the Tuned Lens (Belrose et al), which, maybe with a different end goal, use a similar method. Discussing should be added in the related work section

**Experimental Designs Or Analyses:**

The experimental design is generally sound, with well-structured comparisons between activation communication and natural language-based methods. The use of both synthetic tasks and reasoning benchmarks helps validate the approach across different settings. However, a notable weakness is that all tested models come from the LLaMA family, meaning the study does not evaluate communication between models with different architectures, tokenization schemes, or training distributions. This limits conclusions about the method's general applicability. The comparison with single agent is also a (mild) concern that I have (see more below in the additional questions section).

Additionally, the paper lacks an analysis of why the method work, e.g. inspecting the geometric properties of the activations being communicated. As an example, understanding whether activation similarity between agents correlates with downstream task performance could provide insights into when and why activation-based communication is effective. Metrics like cosine similarity or rank correlation between activations before and after communication could help determine if performance gains are linked to latent space alignment between models. This type of analysis would strengthen the theoretical grounding of the method.

The paper would be a clear accept with tests on a broader range of model families to confirm that activation communication is not specific to Llama's architecture or training paradigms, and additional insights on the effectiveness of the proposed approach.

**Methods And Evaluation Criteria:**

The proposed methods make sense for studying inter-model communication. The use of synthetic datasets is a notable strength. The countries and coordination game allow the authors to isolate the impact of communication without confounding factors. This controlled setup provides clearer insights into how activation-based communication influences performance.

The reasoning benchmarks used are fairly standard in the field, making the evaluation relevant and comparable to prior work.

**Other Comments Or Suggestions:**

In the appendix I found several interesting insights that should be either moved or at least discuss and referenced in the main body of the paper.
For instance,
- the analysis on the interplay between CoT and AC: I would add it or at least discuss it in the main body of the paper. For me, it could be swapped with the compute cost analysis
- the fact that AC over multiple instances of the same model, like NLD, doesn't always outperform the single model setup
- the fact that AC is superior to NLD even with multiple rounds.

For instance, at the end of page 7, when mentioning the in-distribution training of matrix W, there's no reference to the experiment i in section B.3.

This additional information in the main body of the paper would make it even clearer and more informative

**Other Strengths And Weaknesses:**

The paper is clearly written and easy to follow, with well-structured explanations of the motivation, methodology, and experimental results. The key ideas are presented intuitively, making it accessible to both researchers familiar with multi-agent communication and those new to the topic. The figures and tables effectively support the narrative, particularly in illustrating the activation communication process and performance improvements.

**Questions For Authors:**

1. NLD debate is found to improve with increased number of agents (Figure 10 in the paper), do you think your method can compare with that (stronger) setup? I am thinking of instances of your method like f = average or a regularized W matrix learned to transfer across multiple models

**Relation To Broader Scientific Literature:**

The paper builds on prior work in multi-agent LLM communication by replacing costly natural language exchanges with direct activation transfer, reducing compute while improving performance. It extends activation engineering research by using intermediate activations as a communication channel rather than for single-model control. The method is related to model grafting but removes the need for learned routing or fine-tuning, making it more generalizable. Though not fundamental to position the paper, broader comparisons with cross-model embedding alignment techniques would strengthen the work’s positioning, e.g. Relative representations enable zero-shot latent space communication (Moschella et al.)

**Theoretical Claims:**

N/A

---

> ### Author Rebuttal · Authors · 2025-04-01
>
> We thank the reviewer for their feedback, and address their comments below:
>
> 1. > All tested models come from the LLaMA family...
>
> **Please see point #1 in the reply to Reviewer VKxi above.** In summary, we test AC using models **across the LLaMA, Qwen, and Gemma families**, and find that **AC beats NLD across the board, and beats both individual models for 4/5 of the 6 model pairs** on Biographies/GSM8k respectively----demonstrating the efficacy of AC irrespective of model architecture, size, tokenizer, and training data.
>
> 2. > The paper lacks an analysis of why the method work, e.g. inspecting the geometric properties...
>
> First, please see Section 3.3 for a theoretical grounding of this work.
>
> Second, we conduct the following experiment: for each of the 6 pairs of models $A,B$ in the above experiment (see the table from response to Reviewer VKxi), we compute the increase in Biographies performance with AC relative to the average individual performance of $A$ and $B$. We also compute the matrix analog of squared cosine similarity between the models' activation spaces, $||Y^T X||_F^2/(||X||_F^2||Y||_F^2)$ where $X$ is the matrix of $A$'s activations on 3072 sentences from C4 (same dataset used to train $\mathbf{W}$), $Y$ is the same for $B$, and $||\cdot||_F$ is the Frobenius norm. This gives us the following plot (please click link if image not displayed):
>
> https://i.ibb.co/3y6XF7Z7/model-comparison-plot.png
>
> There is a clear positive correlation between similarity of the activation distributions and AC performance gain, as expected; the more aligned $A/B$'s activation spaces, the more semantically meaningful/useful the embedding we graft from $A$ to $B$.
>
> 3. > Though not fundamental to position the paper, broader comparisons...
>
> Thanks for sharing this. We've included a discussion of this and similar papers (e.g., Kornblith19, Similarity of Neural Network Representations Revisited) in the paper.
>
> 4. > A link that I didn't see in the paper...
>
> We thank the reviewer for bringing this to our attention. The "tuned lens" and "logit lens" [1] are related to the theoretical intuition behind this approach that we share in Section 3.3; we have added a discussion of these papers to both Section 3.3 and related works.
>
> 5. > In the appendix I found...
>
> Thanks for pointing this out, we have ensured all experiments in the appendix are either moved to or discussed in the main body of the paper.
>
> 6. > NLD debate is found to improve with increased number of agents...
>
> First, we want to distinguish between **"agents"** (distinct model *instances*) and **"models"** (distinct *LMs*).
>
> With debate, more agents can help because we are sampling different outputs from each agent (model instance), and this can yield diverse reasoning paths that are recombined to produce stronger final outputs.
>
> This isn't true with AC, as one activation grafting step from $A$ to $B$ inherently communicates to $B$ all of $A$'s knowledge/beliefs about the prompt it was given. **We argue this is actually a benefit of AC over NLD, as we don't require increasing token budgets to extract more and more information out of the LMs.**
>
> A similar argument can be made for the number of *rounds* in NLD. Indeed, as shown in Appendix B.5, for 5 of the 7 reasoning benchmarks, **AC beats NLD even with 3-4 rounds while using substantially less compute**, highlighting the superiority and robustness of activations as an alternative “language” for inter-LM communication.
>
> AC could theoretically scale to more than 2 *models*. While letting $f$ be the average function would work out of the box, we saw that direct replacement is much more effective. So, for instance, consider a setup (using the notation from Section 3.1 of our paper) where for any $i, j, k, m$ with $j < k$, we:
>
> 1. run a partial forward pass $B_{\leq j}(x_B)$ to get last-token activation $\mathbf{b}_j$;
> 2. run a partial forward pass $A_{\leq i}(x_A)$ to get $\mathbf{a}_i$;
> 3. replace $\mathbf{b}_j \leftarrow f(\mathbf{a}_i, \mathbf{b}_j)$;
> 4. continue $B$'s forward pass till layer $k$ to get last-token activation $\mathbf{b}_k$;
> 5. run a partial forward pass $C_{\leq m}(x_C)$ to get $\mathbf{c}_m$;
> 6. replace $\mathbf{c}_m \leftarrow f(\mathbf{b}_k, \mathbf{c}_m)$;
> 7. then continue $C$'s forward pass till decoding is complete.
>
> This 3-model setup can extend to an arbitrary number of models. We leave this extension of our approach to future work.
>
> 7. > The paper would be a clear accept with tests on a broader range of model families to confirm that activation communication is not specific to Llama's architecture or training paradigms, and additional insights on the effectiveness of the proposed approach.
>
> **We hope that with the provided responses & additional strong results, you consider raising your score to support clear acceptance. Please let us know if there are any additional questions or concerns; we'd be happy to address them.**
>
> [1] nostalgebraist. Interpreting gpt: the logit lens, 2020.

---

> > ### Comment · Reviewer_EeXt · 2025-04-02
> >
> > I thank the reviewers for they work on the rebuttal.
> >
> > The table you shared in the reply to reviewer VKxi is exactly what I had in mind. It's interesting that GSM8k benefits the most from AC, whereas improvements on Biographies, if any, are marginal. I'm comparing model B vs AC column.
> >
> > > First, please see Section 3.3 for a theoretical grounding of this work.
> >
> > I had read Section 3.3 and do not see any strong theoretical grounding there. I'm not saying this as a weakness, especially given that I find in that section several intuition on why your method work. What I don't find is a formal or empirical evidence that can trace back to why AC performs better. See for instance [1], which came out roughly around the time of my review. I also want to add that a similar intuition was known at least since 2020 with work on self-supervised learning of visual representations.
> >
> > The plot you share on similarity vs performance does not give us much more intuition, but it's a nice preliminary analysis, especially if we consider the short amount of time you had to run it. I suggest its inclusion in the paper together with the table above.
> >
> > > This isn't true with AC, as one activation grafting step from to inherently communicates to all of 's knowledge/beliefs about the prompt it was given.
> >
> > I disagree with this, AC communicates rich representation of one reasoning trajectory of model A, or at least the trajectory up until that point and potenally many different ones from that point onwards. At the same time I agree with you on the increased efficiency of AC over NLD.
> >
> >
> > Finally, I have considered the additional set of experiments and the preliminary analysis and I don't see any reasons why this paper shouldn't be accepted. I am increasing my score to 4.
> >
> >
> > [1] Skean et al, Layer by Layer: Uncovering Hidden Representations in Language Models, 2025

---

> > > ### Author Response · Authors · 2025-04-08
> > >
> > > We thank the reviewer for their response, and appreciate the additional valuable feedback! We will definitely include more extensive discussion of these points in the final paper if accepted.

---

### Official Review · Reviewer_gYwD · 2025-03-13

**Overall Recommendation:** 3

**Summary:**

The paper proposes an alternative method for communication between language models (LMs) that does not rely on natural language. Instead, the authors introduce a technique for LMs to communicate with activations. More specifically, intermediate activations from one model are injected into another's computation at an intermediate layer, allowing models to exchange information in a manner that the authors argue is more efficient than text. Through experiments, this method is shown to improve reasoning performance in multi-agent tasks, outperforming natural language-based communication across various datasets while using less compute as compared to natural language debate. The paper validates the approach through experiments on multi-player coordination games and reasoning benchmarks, demonstrating the method's robustness across different model sizes.

**Claims And Evidence:**

Yes, no issue.

**Essential References Not Discussed:**

NA

**Experimental Designs Or Analyses:**

Yes, no issue.

**Methods And Evaluation Criteria:**

Yes, no issue.

**Other Comments Or Suggestions:**

The work by Phametal.,2024, though different, should also be a method to compare as a baseline to better understand the effect of using intermediate activations versus embeddings in multi-agent frameworks.

There are also highly similar approaches that also rely on using intermediate activations from a language model to inject into intermediate layers of the same language model to improve reasoning (e.g. https://arxiv.org/abs/2412.06769 and https://arxiv.org/abs/2502.05171). These approaches do not need another training stage to train a separate W to map activations from one LM to another. More discussion about these similar works would help the reader better understand the difference.

**Other Strengths And Weaknesses:**

Strengths:
Innovation is timely with the popularity of multi-agent framework
Experimental results show better computational efficiency compared to natural language baseline.
Weaknesses:
Lack of comparison with other similar approaches such as model merging and single model with latent reasoning. These approaches, though not framed as multi-agent approaches, have close similarities in using activations from intermediate layers of an LM injecting into other intermediate layers of the same LM.
The approach needs access to the models’ parameters which is not feasible for state-of-the-art close-source LMs.
Need for model-model pair training of W to map representations from one LM to another LM if architecture is different.

**Questions For Authors:**

NA

**Relation To Broader Scientific Literature:**

The proposed approach contributes to the field of multi-LM agent framework which is of interest to a broad ML community.

**Theoretical Claims:**

NA

---

> ### Author Rebuttal · Authors · 2025-04-01
>
> We thank the reviewer for their feedback, and address their comments below:
>
> 1. > Lack of comparison with other similar approaches such as model merging
>
> To adequately scope our paper, we chose to limit our focus to task-agnostic methods. Model composition/merging methods are extensively discussed in Sections 2 and 3.2, but they require a substantial amount of additional task-specific parameters and data, hence we do not compare to these. Furthermore these methods require much more compute in the form of LM finetuning, layer or router training, etc.; our approach is far more compute efficient.
>
> 2. > The approach needs access to the models’ parameters which is not feasible for state-of-the-art close-source LMs.
>
> Exploring API-only approaches is highly limiting (see AC's meta review of [1] at ICLR last year). Furthermore, recent releases of powerful open-source models [2] merit the development of embedding-based techniques.
>
> 3. > Need for model-model pair training of W to map representations from one LM to another LM if architecture is different.
>
> Note that learning $\mathbf{W}$...
> - needs to happen **exactly once** for each *model pair*
> - introduces zero additional task-specific parameters and data by virtue of requiring only general text, e.g. sequences from $A$ and/or $B$’s pretraining data mixes
> - is quite sample-efficient: as mentioned in the paper, we use $3072$ sentences to train $W\in \mathbb{R}^{4096\times 3072}$, since linear regression with $d$-dimensional input has a sample complexity of $O(d)$ [3]
>
> Furthermore, we find empirically that *even when models have different architectures, tokenization schemes, or training distributions, we do* **not** *need to train a mapping matrix $\mathbf{W}$ to attain SOTA results.*  **Please see point #1 in the reply to Reviewer VKxi above.**
>
> 4. > The work by Phametal.,2024, though different, should also be a method to compare as a baseline to better understand the effect of using intermediate activations versus embeddings in multi-agent frameworks.
>
> Pham24 propose communicating the *input (tokenizer) embeddings* between models, meaning the two models **must have the same tokenizer and embedding table** to even run their approach. This severely limits the applicability of their method; in particular, given that all our experiments use model pairs (e.g., LLaMA-3.2-3B and LLaMA-3.1-8B) with distinct tokenizers and/or embedding layers, we unfortunately cannot compare against Pham24. Also, note Pham24's approach still faces substantial information loss relative to the model activations and, more importantly, **does not save compute**, as the number of embeddings passed between models is *the same as the number of tokens passed between models in natural language communication*.
>
> 5. > There are also highly similar approaches that also rely on using intermediate activations from a language model to inject into intermediate layers of the same language model to improve reasoning (e.g. https://arxiv.org/abs/2412.06769 and https://arxiv.org/abs/2502.05171). These approaches do not need another training stage to train a separate W to map activations from one LM to another. More discussion about these similar works would help the reader better understand the difference.
>
> Thank you for raising this point; we have added discussion of these approaches to our paper. In summary, such latent reasoning approaches involve spending extra compute by doing "CoT in activation space," e.g. by grafting LM activations into other layers/later forward passes through the same model; our approach can be viewed as doing exactly the same thing, but instead "outsourcing" the CoT to another model (and thus reaping benefits from greater diversity of thoughts/reasoning paths from distinct models).
>
> Also, note that as shown above, we find that even when models have different architectures, tokenization schemes, or training distributions, we do *not* need to train a mapping matrix $\mathbf{W}$ to attain SOTA results.
>
> **We hope that with the provided responses & additional strong results, you consider raising your score to support clear acceptance. Please let us know if there are any additional questions or concerns; we'd be happy to address them.**
>
> [1] Chau Pham, Boyi Liu, Yingxiang Yang, Zhengyu Chen, Tianyi Liu, Jianbo Yuan, Bryan A. Plummer, Zhaoran Wang, and Hongxia Yang. Let models speak ciphers: Multiagent debate through embeddings, 2024.
>
> [2] Abhimanyu Dubey et al. The llama 3 herd of models, 2024.
>
> [3] Vapnik, V. N. An overview of statistical learning theory, 1999.

---

### Official Review · Reviewer_VKxi · 2025-03-14

**Overall Recommendation:** 2

**Summary:**

The paper considers this fundamental question "as LLMs are increasingly capable of handling larger, more complex tasks (sometimes with “super-human” ability), might they communicate more effectively in representations of higher dimension than natural language?". It proposed a simple technique where LMs communicate via activations.

**Claims And Evidence:**

The primary claim in the paper is that LMs can communicate using activations. An LM B’s computation is paused at an intermediate layer, then its current activation is combined with another LM A’s intermediate activation via some function f, and then f’s output is passed into the next layer of B and the forward pass is continued till decoding is complete.  The experimental evaluation is done over 7 reasoning benchmarks and 2 multiplayer games. The activation communication protocol exhibits up to 27.0 improvement over natural language communication across these datasets using <1/4 the compute.

**Essential References Not Discussed:**

Yes, references are discussed. The paper demonstrates good familiarity with literature.

**Experimental Designs Or Analyses:**

The paper considers rather simple datasets (2 multiplayer and 7 reasoning tasks) and two models (llama 3b and 8b).

**Methods And Evaluation Criteria:**

The presented method is rather simplistic but the experimental evaluation demonstrates its value.

**Other Comments Or Suggestions:**

It would be useful to expand the empirical evaluation to more and diverse datasets.

**Other Strengths And Weaknesses:**

A rather simple idea of using activations is shown to have good empirical value on the considered experiments.

The weights learned for projecting activation from one model to another is task-independent.

The experimental evaluation can be improved with better baselines than single model and NL debate.

**Questions For Authors:**

Why was k = j = 26 selected as the layer?

The explanation in line 25-255 is not convincing about why the experimental comparison with Pham24 was avoided.  Could you please elaborate further?

**Relation To Broader Scientific Literature:**

The paper is an interesting contribution to multi-LLM inference.

**Theoretical Claims:**

There are no significant theoretical claims in the paper.

---

> ### Author Rebuttal · Authors · 2025-04-01
>
> We thank the reviewer for their feedback, and address their comments below:
>
> 1. > The paper considers rather simple datasets (2 multiplayer and 7 reasoning tasks) and two models (llama 3b and 8b).
>
> In Appendix B.6, we display results on **the entire MMLU benchmark (57 datasets)**, spanning various domains and difficulty levels. We find that **AC matches/outperforms NLD on 48/57 datasets**, demonstrating our approach's value.
>
> While our initial experiments were only with LLaMA models, note: (1) These LMs are among *SOTA open-source models*, meriting our focus; and (2) we extensively vary both the LLaMA suite (LLaMA-2, 3, 3.1, 3.2) and parameter count (1-70B) as shown in Figure 3 of the paper. This is already a broad coverage of models/model sizes.
>
> **However, we share additional results using models from the Qwen-2.5 and Gemma-2 families below.** Each cell contains two results: Biographies score / GSM8k score.
>
> | Model Pair ($A,B$)                                           | $A$                              | $B$                              | NLD                              | AC                                        |
> | ------------------------------------------------ | -------------------------------- | -------------------------------- | -------------------------------- | ----------------------------------------- |
> | LLaMA-3.2-3B, LLaMA-3.1-8B  | $79.4\pm0.0/58.0\pm4.9$          | $83.9\pm0.0/60.0\pm4.9$ | $80.2\pm0.1/\mathbf{75.0\pm4.3}$ | $\mathbf{84.6\pm0.0}/64.0\pm4.8$                   |
> | Qwen-2.5-1.5B, Qwen-2.5-3B  | $59.4\pm0.9/20.0\pm0.9$          | $85.5\pm1.1/35.0\pm1.1$          | $63.2\pm1.1/65.0\pm1.1$          | $\mathbf{89.6\pm1.0}/\mathbf{70.0\pm1.0}$ |
> | Gemma-2-2B Gemma-2-9B      | $83.0\pm1.1/45.0\pm1.1$          | $\mathbf{94.6\pm0.9}/80.0\pm0.9$ | $70.3\pm1.0/70.0\pm1.0$          | $88.1\pm0.7/\mathbf{90.0\pm0.7}$          |
> | Qwen-2.5-1.5B, LLaMA-3.2-3B | $59.4\pm0.9/20.0\pm0.9$          | $79.4\pm0.0/58.0\pm4.9$          | $75.4\pm1.0/75.0\pm1.0$          | $\mathbf{79.5\pm1.0}/\mathbf{75.0\pm1.0}$          |
> | LLaMA-3.2-3B, Gemma-2-2B    | $79.4\pm0.0/58.0\pm4.9$ | $83.0\pm1.1/45.0\pm1.1$          | $62.5\pm1.1/55.0\pm1.1$          | $\mathbf{84.0\pm0.1}/\mathbf{60.0\pm1.1}$          |
> | Qwen-2.5-1.5B, Gemma-2-2B   | $59.4\pm0.9/20.0\pm0.9$ | $\mathbf{83.0\pm1.1}/45.0\pm1.1$            | $49.3\pm1.1/50.0\pm1.1$          | $73.0\pm1.1/\mathbf{55.0\pm1.1}$          |
>
> Note the following:
> - **AC beats NLD across the board**, and **beats both individual models for 4/5 of the 6 model pairs** on Biographies/GSM8k respectively----demonstrating the efficacy of AC irrespective of model architecture, size, tokenizer, and training data
> - These results are obtained *without* training $\mathbf{W}$, meaning we do *not* need to train a projection layer between activation spaces to attain SOTA results, even for extremely distinct models! (We hypothesize this is because we are only replacing $B$'s *last*-token activation, hence $B$ can learn from $A$ without an extreme alteration to its activation distribution)
>
> 2. > The experimental evaluation can be improved with better baselines than single model and NL debate.
>
> We limit our focus to task-agnostic methods; existing model composition/grafting methods require substantial task-specific parameters/data & much more compute in the form of LM finetuning, layer or router training, etc., hence we do not compare to these.
>
> Regarding multiagent debate, the NLD setup that we evaluate against is the predominant method of NL communication. This is quite a strong/flexible NL approach, involving CoT and allowing varied numbers of agents/rounds. (In fact, as shown in Appendix B.5, we find **AC outperforms NLD even with 3-4 rounds of debate on 5 of the 7 reasoning datasets**.)
>
> 3. > Why was k = j = 26 selected?
>
> See Section 3.3, lines 201-222. This reasoning is precisely why we choose a layer around halfway through the LM; indeed, Hernandez24 find that by around the halfway point of an LM's computation, it has developed "enriched entity representations" of the input that would be quite useful for communication compared to the next-token representations of later layers.
>
> We validate this empirically; Figure 2 shows 2D contour plots of accuracy over different $k,j$ values. For computational reasons we only do this hyperparameter sweep on the Countries and Tip Sheets datasets and simply cross-apply the optimal values here, $k=j=26$, to the reasoning benchmarks and find that the values seem to generalize well across datasets, which is quite interesting in its own right.
>
> 5. > The explanation in line 25-255 is not convincing about why the experimental comparison with Pham24 was avoided. Could you please elaborate further?
>
> **Please see point #4 in the reply to Reviewer gYwD below.**
>
> **We hope that with the provided responses & additional strong results, you consider raising your score to support clear acceptance. We are happy to address any additional questions.**

---

### Decision · Program_Chairs · 2025-05-01

**Decision:**

Accept (poster)

**Comment:**

The core innovation of this work is to question whether natural language is the best communication channel for LM to LM information sharing. The implementation of this idea is to have activations from an intermediate stage of the first model be combined (via projection) with those of the second model which then continues computation. This softer channel is than evaluated on multiplay and reasoning settings.  The insight also begs the question as to what defines a “model”, is this two models or just a single one with a specific control flow. This therefore warrants evaluation via a single model.  These types of changes also beg questions about specific design choices (layer, projection learning, etc) which are not required when using NL.

Many of these concerns (benchmarks, models, and design) are resolved in the author response.